# High-resolution differentiation and identification of the clinically relevant pathogens *Haemophilus influenzae* and *Haemophilus aegyptius*: combined whole-cell MALDI-TOF MS and nano-LC-MS/MS proteotyping for defining protein biomarkers

Hamideh Hamidi,[1,2] Guillem Seguí,[3,4] Alexandra Gillberg,[3] Alireza Ghassempour,[2] Edward R. B. Moore,[1,3,4] Roger Karlsson[1,3,4]

**ABSTRACT** Precise differentiation between closely related species is crucial for effective clinical diagnosis and treatment. *Haemophilus influenzae* and *Haemophilus aegyptius* are phenotypically similar but differ in pathogenic potential, with *H. influenzae* involved in respiratory and invasive infections, and *H. aegyptius* mainly associated with ocular disease and, rarely, severe systemic outcomes. Traditional phenotypic methods and gene-based analyses often struggle to reliably distinguish them due to high sequence similarity and shifting taxonomic boundaries. In this study, we present a high-resolution proteotyping method that combines whole-cell MALDI-TOF MS, nano-LC-MS/MS, and comparative genomics to identify and verify species-specific protein biomarkers across a broad mass range (*m/z* 3,000–40,000). Using sequenced reference strains and extensive *in silico* analyses, we found 31 unique protein biomarkers that allow reliable differentiation between *H. influenzae* and *H. aegyptius*. These markers were confirmed through LC-MS/MS peptide mapping and comparative genomics, showing consistent sequence differences and phylogenetic clustering. Our results demonstrate that combining advanced proteomic profiling with genome-based validation effectively addresses the limitations of conventional identification methods. This approach enhances diagnostic accuracy for closely related pathogens and facilitates the ongoing refinement of spectral and genomic reference databases, which are essential for precise microbial identification in clinical microbiology.

**IMPORTANCE** Species-level resolution of clinically relevant pathogens remains a challenge in clinical diagnostics. *Haemophilus influenzae* and *Haemophilus aegyptius* are particularly problematic because of their high genetic similarity, overlapping phenotypes, and distinct clinical implications. Conventional techniques often fail to reliably distinguish them, leading to frequent misclassification. Although diagnostic issues within the *H. influenzae* complex are well recognized, differentiation between *H. influenzae* and *H. aegyptius* remains underexplored. Accurate and rapid identification is essential, as these opportunistic pathogens differ in disease manifestations and epidemiology, and misidentification can result in suboptimal treatment. Our study presents a robust high-resolution nano-LC-MS/MS proteotyping approach, combined with whole-cell MALDI-TOF MS and comparative genomics, to confidently classify *H. influenzae* and *H. aegyptius* at species and strain levels. By identifying species-unique peptide biomarkers, this method fills a diagnostic gap and improves species-level microbial diagnostics, understanding of pathogenicity, and epidemiological surveillance.

**KEYWORDS** proteomics, genomics, MALDI-TOF, mass spectrometry, biomarkers, *Haemophilus* species, *H. influenzae*, *H. aegyptius*

Address correspondence to Roger Karlsson, roger.karlsson@gu.se.

Hamideh Hamidi and Guillem Segui contributed equally to this article. Hamideh Hamidi performed all experiments and spectral validations; Guillem Segui performed biomarker validation by genomics, data mining, and bioinformatics tools.

R.K. is affiliated with Nanoxis Consulting AB, Gothenburg, Sweden. The company did not have any influence on the collection, analysis, or interpretation of the data, the writing of the paper, or the decision to submit for publication.

See the funding table on p. 19.

In recent years, matrix-assisted laser desorption/ionization time-of-flight mass spectrometry (MALDI-TOF MS) has revolutionized microbiological identification of infectious bacteria from clinical samples (1–6). MALDI-TOF MS is primarily used for rapid and reliable species-level identification by comparing experimental mass-to-charge (*m/z*) profiles of intact, low-molecular-weight bacterial proteins to known reference spectra. However, MALDI-TOF MS has limitations in differentiating specific bacterial pathogens from closely related species with highly similar spectral profiles. To accurately distinguish difficult-to-identify taxa, the detection of specific biomarkers can be combined with spectral profile analysis, utilizing taxonomically significant MS peaks to generate artificial spectra (7). For example, *Francisella tularensis* subspecies can be characterized by identifying specific biomarkers through MALDI-TOF MS and comparative proteogenomic analysis (8). This approach can also be used to identify bacterial strains in environmental samples, especially when databases lack appropriate reference spectra (9, 10).

*Haemophilus* species are gram-negative bacteria that commonly inhabit the respiratory tracts and mucosal surfaces of humans and animals. The *Haemophilus* genus includes fourteen recognized and validly published species, with *H. influenzae* being the type and most clinically significant species of the genus (10). *Haemophilus influenzae* and *Haemophilus aegyptius* are frequently encountered in clinical microbiology laboratories and exhibit a range of pathogenicity, from non-pathogenic commensals to highly pathogenic strains responsible for severe conditions such as respiratory tract infections (RTIs), conjunctivitis, and life-threatening invasive diseases like septicemia, sepsis, and meningitis. Various protocols have been developed to distinguish *H. influenzae* from closely related species of lesser pathogenic importance. However, accurate identification based on phenotypic characteristics remains challenging and often unreliable (11). DNA-DNA hybridization analysis suggests that *H. influenzae* and *H. aegyptius* may not be distinct species (12), and MLSA places them within the core of the species (13, 14). Despite this, *H. influenzae* and *H. aegyptius* have traditionally been classified as distinct species. *H. aegyptius* is mainly associated with conjunctivitis and other ocular infections. Clinical diagnosis typically depends on observing symptoms such as eye redness and discharge, which are confirmed through laboratory testing. In some cases, infections caused by *H. aegyptius* have been linked to more serious conditions, like purpuric fever, which can be fatal (15). Conversely, *H. influenzae* is associated with a variety of infections, including respiratory tract infections, meningitis, and epiglottitis. The differing clinical presentations of these two species underscore the importance of accurate identification to guide appropriate treatment and management.

MALDI-TOF MS can reliably identify species of *Haemophilus* quickly, easily, and at low cost (16, 17), although, using standard protocols, some taxonomically close species cannot be distinguished, even with high-quality spectra. This applies to *H. influenzae* and related species, which exhibit significant intraspecies variability (5, 18, 19). MALDI-TOF MS analysis has previously demonstrated the close relationship between *H. influenzae* and *H. aegyptius*, as well as other related species (17, 19–22). Therefore, there is a need to develop protocols that enable more accurate differentiation of closely related species or subspecies, particularly for diagnostic purposes.

The use of tandem MS analysis of tryptic peptides from extracted proteins has proven to be a useful alternative for classifying and identifying microorganisms through bottom-up proteotyping (23, 24). When employing "bottom-up" approaches, interpreting peptide sequences effectively depends on the bioinformatics pipeline and the appropriate protein sequence database; therefore, proteomics heavily relies on genomics information. An effective strategy is to focus on intact proteins using "top-down" proteomics, as shown in whole-cell MALDI-TOF MS analysis (25); however, in some cases, further taxonomic identification may require characterization of discriminative protein biomarkers (26). To improve MALDI-TOF MS-based databases and increase species differentiation, the use of tandem mass spectrometry-based proteomics has been employed (27–30), for instance, in the discovery of biomarkers of *Neisseria meningitidis* (30). The goal of this study was to combine shotgun nano-LC-MS/MS proteotyping

with detailed genomics investigations to identify reliable MALDI-TOF MS biomarkers for difficult-to-identify *Haemophilus* species, focusing on *H. influenzae* and *H. aegyptius*.

To address this, a high-throughput MALDI-TOF MS protocol was first optimized, enabling the acquisition of high-quality mass spectra within an extended molecular mass range of 3,000–40,000 Da. Second, informative and reproducible MALDI-TOF-MS spectra were obtained under defined conditions for well-characterized strains closely related to *H. influenzae* and *H. aegyptius*. Third, bottom-up tandem mass spectrometry was used to identify novel species-specific peptides and key proteins in the MALDI-TOF MS biomarker spectra. Finally, comprehensive comparative genomics of all strains from the two species available at the NCBI was employed to validate the candidate biomarkers. These biomarkers significantly enhanced the identification of *H. influenzae* and *H. aegyptius* isolates using MALDI-TOF MS.

## MATERIALS AND METHODS

### Bacterial strains and cultivation

For this study, nine bacterial strains were acquired from the Culture Collection University of Gothenburg (CCUG) (https://www.ccug.se/): *H. influenzae* (CCUG 23945$^T$, CCUG 23969, CCUG 26214, CCUG 29539, CCUG 33775, and CCUG 72045), and *H. aegyptius* (CCUG 628, CCUG 25716$^T$, and CCUG 26840) (Table 1). Cultures were grown on Chocolate and Blood Agar medium (5% horse blood; Substrate Dept., Bacteriology Laboratory, Sahlgrenska University Hospital) with 5% $CO_2$ and incubated at 37°C for 18 ± 2 h, before whole-cell MALDI-TOF MS analyses. The strains of *Haemophilus* spp. included in this study, isolated from patients in South-West Sweden, have been identified at the species level, according to routine laboratory protocols. The selected strains were screened by 16S rRNA gene sequence analysis. Comparative genomic BLAST analyses were performed against type strain sequence databases on https://blast.ncbi.nlm.nih.gov and also EzBioCloud's 16SrRNA database. All bacterial cultures and sample preparation have been done in a BSL2 laboratory.

### Chemicals and matrix solutions

Alpha-cyano-4-hydroxycinnamic acid (CHCA), 2,5-dihydroxybenzoic acid (DHB), sinapinic acid (SA), and protein standard II (Part-No. 8207234) were supplied by Bruker Daltonics (Bremen, Germany). Trifluoroacetic acid (TFA), formic acid (FA), absolute ethanol (HPLC grade), acetonitrile (ACN, hypergrade for LC-MS), and water (hypergrade for LC-MS) were acquired from Merck (Darmstadt, Germany). Primarily, a solution of 50% acetonitrile and 0.6% TFA (500 μL acetonitrile, 494 μL water, and 6 μL pure TFA) designated as "TA50 stock solution" was prepared. Five milligrams of SA were dissolved in 200 μL TA50 stock solution (final concentration: 25 mg /mL) to obtain matrix solution "B" for the following experiments. matrix solution "A" (44 mg/mL$^{-1}$ of SA) was prepared in absolute ethanol. CHCA and DHB (10.0 mg/mL$^{-1}$) were also prepared in 50% acetonitrile and 2.5% trifluoroacetic acid.

### MALDI-TOF mass spectrometry

To ensure a high level of standardization and optimal mass patterns, several steps were refined. First, the type and concentration of the matrix solution were investigated; therefore, CHCA, DHB, and SA were tested to produce the best MALDI-TOF MS signals. In the next step, the matrix solvent mixture and the concentration of acid added to the matrix solution were optimized. Then, the sample preparation process was refined by comparing it to the common MALDI-TOF MS protocol. This was done by applying microbial biomass either as individual colonies directly spotted on the target plate as a thin layer or as a cell-free extract containing bacterial proteins in water, acetonitrile, and formic acid. Finally, to ensure reliable taxonomic species identification, the laser energy

**TABLE 1** *Haemophilus* strains analyzed in this study[a]

| Species | Strain Id | Clinical source | Capsule | Variant | Sample origin |
|---|---|---|---|---|---|
| *H. influenzae* | CCUG 23945[T] | Human sputum | No capsule | Biotype II | USA |
| *H. influenzae* | CCUG 23969 | Human sputum | Nd | Nd | Sweden |
| *H. influenzae* | CCUG 26214 | Human sputum | Nd | Nd | USA |
| *H. influenzae* | CCUG 29539 | Human lung abscess | Nd | Nd | USA |
| *H. influenzae* | CCUG 33775 | Human | Type b | Biotype I | USA |
| *H. aegyptius* | CCUG 628 | Nd | Nd | Nd | USA |
| *H. aegyptius* | CCUG 25716[T] | Human conjunctiva | Nd | Nd | USA |
| *H. aegyptius* | CCUG 26840 | Human conjunctiva | Nd | Nd | USA |

[a]Nd, not determined; [T], type strain.

and the number of laser shots were optimized, and calibration constants were adjusted. All of these optimizations are detailed in Supplemental material.

## Sample preparation

The simplest sample preparation method involved directly applying small amounts of biological material (colony biomass) to the MALDI target plate. A double-layer matrix/sample deposition technique was used, utilizing matrix solutions "A" and "B," as previously described. In this process, 0.5 µL of matrix solution "A" was spotted onto a MALDI target. Then, a single bacterial colony was smeared onto the target into the matrix solution "A," and 1 µL of matrix solution "B" was immediately added to enhance cell lysis. The prepared target was then allowed to dry at room temperature before conducting MALDI-TOF MS measurements using the MALDI-TOF MS Biotyper Microflex LT System and the SMART MBT software and reference library for research use only (Bruker Daltonics, Germany). For each strain, MALDI-TOF MS spectra were obtained from four different biological samples (biological replicates), with each sample measured six times (technical replicates). Bruker Protein Standard II (Bruker Daltonics, Part-No. 8207234) and a two-layer matrix deposition method were used for system calibration and to determine the optimal system settings.

## MALDI-TOF MS analysis

Raw spectra from reference *Haemophilus* strains were obtained using the MALDI-TOF MS Biotyper Microflex LT System instrument in linear positive ion mode, covering a mass range from *m/z* 3,000 to 40,000, according to the manufacturer's recommended specifications. The initial laser power was set to 65% when the signal first emerged from the background and the maximum laser power was increased by 2%–5% (e.g., to 70%). Spectra for each replicate were generated by accumulating 2,000 laser shots over 20 target sites, using AutoXecute acquisition software (FlexControl version 3.4, Bruker Daltonics, Germany). Independent spectra were acquired by utilizing different raster layers at individual spots. Mass calibration of whole cells of *H. influenzae* and *H. aegyptius* was performed through external calibration using Bruker Protein Standard II ([M + H]+, *m/z* 23,982.00; [M + H]2+, *m/z* 11,983.00; and other peaks), followed by self-calibration with moderately strong peaks assigned to ribosomal subunit proteins as internal references (Table S1). Ultimately, the most intense *m/z* peaks with a presence greater than 75% were selected for biomarker analysis. The raw data were converted to mzXML format using the Bruker CompassXport utility, which enabled import into the SARAMIS (bioMérieux, Marcy-l'Étoile, France) software environment. The rationale for using SARAMIS was not primary identification, but rather its strength in biomarker-based analysis and categorization of discriminatory peaks. Specifically, SARAMIS was used to identify and classify biomarker peaks into three categories: (i) biomarkers conserved at the genus or higher taxonomic level, (ii) biomarkers specific to either *H. influenzae* or *H. aegyptius*, and (iii) biomarkers shared by all strains of both species.

## Tandem mass spectrometry and shotgun proteomics

As described previously (31), the biomasses of *H. influenzae* (CCUG 23945[T]) and *H. aegyptius* (CCUG 25716[T]) were collected from fresh cultures and suspended in phosphate-buffered saline (PBS) and centrifuged. The resulting pellet was washed three times with 1.0 mL PBS, followed by centrifugation (5 min; 10,000 $\times$ *g*). The cells were resuspended in 150 μL of PBS, to an $OD_{600}$ of 1.0, corresponding to ~$10^8$ bacteria, and lysed by bead-beating, using a TissueLyser (Qiagen, cat. No. 85220, Hilden, Germany), with approximately 50 μL of acid-washed glass beads (size 150–212 μm, cat. No. G1145, Sigma-Aldrich, Buchs, Switzerland) for 5 min at a frequency of 1/25 s. The cell lysates were stored at −20°C until analysis.

## Digestion of samples for proteotyping into peptides

Samples were thawed, and sodium deoxycholate (SDC 5%) was added to a final concentration of 1% (wt/vol). Trypsin (2 μg/mL$^{-1}$ 20 mM ammonium bicarbonate, pH 8) was added, and the samples were digested for 8 h at 37°C. The tryptic peptides were recovered by acidification with 10% formic acid, precipitated SDC was removed, and the supernatant was stored at −20°C until analysis.

## Nano-LC-MS/MS analysis of proteotyping samples

Nano-LC-MS/MS measurements were performed using a Q Exactive HF mass spectrometer (ThermoFisher Scientific, Germany) interfaced with an Easy nLC 1200 liquid chromatography system (ThermoFisher), after desalting and concentrating the peptides, using PepClean C18 spin columns (ThermoFisher), according to the manufacturer's guidelines. Peptides were trapped on an Acclaim Pepmap 100 C18 trap column (100 μm $\times$ 2 cm, particle size 5 μm, ThermoFisher) and then separated on an in-house packed analytical column (75 μm $\times$ 300 mm, particle size 3 μm, Reprosil-Pur C18, Dr. Maisch, Ammerbuch, Germany), using a linear gradient from 7 to 35 percent B (acetonitrile in 0.2% formic acid) over 45 or 75 min, followed by an increase to 100 percent B for 5 min. The MS/MS analysis was carried out in a data-dependent mode, and the top 10 most intense precursor ions, with charge states of 2–4, were fragmented. The precursor ion mass spectra were acquired at a resolution of 60,000, *m/z* 400–1,600. The isolation window was set to *m/z* 1.2. MS2 spectra were recorded at a resolution of 30,000, *m/z* 200–2,000. The parameters for dynamic exclusion were set to 20 s and 10 ppm.

## MS/MS database search

The MS/MS spectra were searched against a protein database that included all proteins annotated in the RefSeq genome sequences database (retrieved August 2022) for *H. influenzae* (CCUG 23945[T]) and *H. aegyptius* (CCUG 25716[T]). MS/MS spectra were assigned with the following settings: maximum missed cleavages allowed were 2; mass tolerance on the parent ion was 5 ppm; and MS/MS mass tolerance was ±0.1 Da. Carbamidomethylated cysteine residues (+57.0215) were set as the static modification, and oxidation of methionine was set as the variable modification. The protein identifications were validated based on the presence of at least two identified peptides. Additionally, proteins detected with only a single peptide were considered if their molecular weight was below 12 kDa.

## Data evaluation

Theoretical masses of each protein were calculated for both single-charged and double-charged proteins, using a Compute pl/Mw tool on the ExPASy (Expert Protein Analysis System) proteomics server (https://web.expasy.org/compute_pi/) of the Swiss Institute of Bioinformatics (32) from translated amino acid sequences in FASTA format. The amino acid sequences were subsequently evaluated to predict lost "N-Met" (−131.0 Da) and N-terminal acetylation (+42.0 Da). Post-translated N-terminal methionine loss was considered, based on the "penultimate" amino acid type.

The theoretical *m/z* values were subjected to the following freely accessible web-based application, SPECLUST (http://co.bmc.lu.se/speclust/common.pl), and compared with consensus peak lists for each strain of *H. influenzae* and *H. aegyptius*, using the "peaks in common" option with "Width in peak match score 1.0 Da."

The BLAST function of the NCBI (https://blast.ncbi.nlm.nih.gov/Blast.cgi) was executed against type strain sequence databases to perform protein-vs-translated DNA Blast searches for *H. influenzae* and *H. aegyptius*.

## Bioinformatics analysis, genome retrieval, and comparative genomics

A total of 892 genomes were retrieved from the NCBI RefSeq database (33) for comparative analysis, comprising 863 genomes classified as *H. influenzae*, 9 *H. influenzae* biotype *aegyptius*, 6 *H. aegyptius*, and 14 *Haemophilus* spp. Given the large size of the data set, comparisons were performed in groups of 100 genomes, against the reference strains, namely *H. influenzae* NCTC 8143$^T$ (=CCUG 23945$^T$) and *H. aegyptius* NCTC 8502$^T$ (=CCUG 25716$^T$), and pairwise average nucleotide identity (ANI) was calculated using the pyANI 0.2.11 (34) to assess genomic similarity.

The genomes were clustered according to their ANI reference type strains' similarity profile. A subset of 93 genomes, comprising 17 strains closely related to the type strain of *H. aegyptius* and 76 strains related to the type strain of *H. influenzae,* was selected as representative genomes for the genomic validation of the protein candidate biomarkers.

## Genome annotation and biomarker analysis

The 93 selected genomes were annotated by Prokka v1.14.6 (35) to generate a standardized gene prediction. A subsequent analysis was focused on the potential biomarkers identified in this study. The presence of these proteins across the sequences of all 93 genomes was assessed and extracted, using BLASTp v2.15.0 (36) with an E-value threshold of 100, to ensure a comprehensive analysis for each biomarker within each genome.

## Phylogenetic analysis

The candidate protein biomarkers were evaluated through phylogenetic analysis to confirm if protein sequences separated *H. influenzae* and *H. aegyptius* into distinct species. Protein sequences were aligned using ClustalW v2.1 (37), and multi-locus sequence analyses (MLSA) were performed on concatenated biomarkers and a subset excluding non-variable biomarkers. Phylogenetic trees were constructed using the neighbor-joining method with a bootstrap value of 1,000.

## RESULTS AND DISCUSSION

In this study, we combined high-resolution nano-LC-MS/MS proteotyping with whole-cell MALDI-TOF MS and comparative genomics to identify 31 distinct protein biomarkers uniquely and consistently associated with *H. influenzae* and *H. aegyptius*. These biomarkers demonstrate strong potential for accurate species-level identification. Genomic validation further confirmed their specificity and reliability as diagnostic markers.

A simple and reliable method was developed to identify well-resolved MALDI-TOF MS peaks within the range of *m/z* 3,000–40,000, using whole-cell MALDI-TOF MS of bacterial strains with sequenced genomes. The approach involves three distinct steps: measuring whole-cell MALDI-TOF MS mass spectra, identifying proteins present in the bacterial extract through nanoLC-MS/MS shotgun analysis, and finally constructing a database by comparing the experimentally observed mass-to-charge (*m/z*) values of selected biomarkers with proteins detected by nano-LC-MS/MS. This database has been employed as a typing method for difficult-to-identify taxa. This was especially important to avoid false peak assignments caused by numerous possible post-translational modifications and incorrect annotations. The focus was on identifying proteins

with molecular weights within the *m/z* 3,000–40,000 range to significantly increase the information content of the mass spectra, enabling the reproducible detection of enough biomarkers with specificities at the closely related species level. To demonstrate the approach, *H. influenzae* (CCUG 23945[T]) and *H. aegyptius* (CCUG 25716[T]) were used as reference strains, and the expected *m/z* values for other sequenced *H. influenzae* and *H. aegyptius* strains were predicted. These predictions were experimentally verified as *H. influenzae* and *H. aegyptius* once several biomarkers were established for them.

## Optimization of analytical procedures

The simplest processes that could be automated to produce MS data with high information content were evaluated, including the type and concentration of the matrix, sample preparation methods, the solvent mixture for dissolving the matrix, the amount of acid added, and measurement variables like laser energy and the total number of shots per sample (Supplemental material).

## Whole-cell MALDI-TOF MS biomarkers for *H. influenzae* and close relatives

For eight representative strains of *H. influenzae* and *H. aegyptius*, 192 whole-cell MALDI-TOF MS spectra were collected. The resulting mass spectra showed subtle differences but enabled the detection of small mass shifts in the peaks of the ribosomal protein spectra. At least 20 optimal mass differences from a single strain over the mass range of *m/z* 3,000–40,000 were considered. The reference for each strain, known as the main spectrum (MSP), was generated using Bruker Biotyper software, which included the 150 most prominent peaks with peak occurrence frequencies of at least 75%. SARAMIS software was used to identify biomarker peaks and categorize them. These categories included biomarkers that identified strains only to the genus or higher taxonomic levels (category I), biomarkers unique to either *H. influenzae* or *H. aegyptius* (category II), and biomarkers found in all strains of both species (category III). Any other peaks showing variability within taxa, such as those likely strain-specific or with variable expression, were excluded from further analysis.

To enable more accurate determination of *m/z* peaks, especially when considering internal calibration, identifying the proteins responsible for these peaks in MALDI-TOF MS spectra is crucial. For this approach, protein standard II, which contains three proteins (trypsinogen, protein A, and bovine serum albumin—BSA), was applied to the smeared cells at a final concentration of 50 ng/μL. This concentration has been shown to produce six reliable, reproducible calibrant peaks as mono- and double-charged ions within the *m/z* 10,000–40,000 range. Additionally, ensuring uniform distribution of the internal calibrants throughout the sample is important. Careful optimization of sample preparation, data acquisition settings, and post-acquisition analysis is necessary to achieve the most accurate and reliable results. First, the MALDI-TOF MS spectra were calibrated using the six standard calibrant *m/z* values mentioned above. The accuracy of the remaining calibrant peak was then evaluated. The masses of stable peaks attributed to ribosomal subunit proteins were identified, allowing their use for internal recalibration. Analysis of a single MALDI-TOF MS spectrum showed the accurate mass within an error range of 5.51–265.93 ppm, corresponding to molecular weights from 10,000 to 40,000 Da. The *m/z* values, average intensities, and peak occurrence frequencies for eight strains of *H. influenzae* and *H. aegyptius* were determined through multiple measurements (Table S2).

The mass tolerance window used for matching well-resolved peaks observed in the spectra of different strains was 1 Da within the molecular mass range of *m/z* 2,000–20,000. Tentative assignments were made for larger and low-intensity protein peaks using a wider mass tolerance window of 5 Da.

## Proteome of *H. influenzae* and *H. aegyptius* type strains determined by nano-LC-MS/MS shotgun analysis

Shotgun proteomics of protein extracts from bacterial cultures was employed to enhance and ensure consistent protein detection. The two most closely related species

of the genus *Haemophilus* (i.e., *H. influenzae* and *H. aegyptius*) served as a model system to generate the LC-MS/MS database. Each replicate was analyzed separately. In total, approximately $4 \times 10^3$ non-redundant peptides were identified from the strains studied, with an average of around 11,800 protein groups per strain (Tables S3 and S4). Proteins with at least two identified peptides, including those detected with a single peptide and a molecular weight below 10 kDa, were considered. Our final list of proteins contained 500 entries after filtering out basic proteins with a molecular weight under 40 kDa.

## Identification of proteins contributing to *m/z* signals in *H. influenzae* and *H. aegyptius* whole-cell MALDI-TOF MS spectra

Each *m/z* observed in the whole-cell MALDI-TOF MS analysis was compared to the *m/z* signals of candidate biomarkers identified through nanoLC-MS/MS, with a focus on ribosomal proteins. This method helped prevent false peak assignments by considering entire proteins and their possible methionine removal and acetylation in single- and double-charged forms. The 101 peaks in the MALDI-TOF MS spectra could be assigned to 51 ribosomal proteins and other structural proteins, such as HU family DNA-binding proteins. Most proteins are detectable in both single- and double-charged forms, while post-translational modifications (PTMs), such as the loss of initiator methionine, are considered following the N-end rule. Table 2 reports different protein isoforms along with their post-translational modifications in ribosomal proteins: S11 (A → R), S13, S3 (N → S), S4 (T → A), RS5 (S → A), S7 (N → K), S9 (E → X), L10 (G → D, A → R, V → A), L11 (S → T), L13 (A → V, G → A), L21 (A → E, X → V), L24 (A → P), L25, L27 (S → N, S → V), L30 (M → I), L6 (V → G), L9 (T → A), L7/12, ribosomal RNA large subunit methyltransferase (S → F, G → S), and DNA-binding protein HU (D → N, E → K), mapped to *H. influenzae* CCUG 23945[T] and *H. aegyptius* CCUG 25716[T]. As shown in Table 2, the ribosomal proteins S14, S15, S16, S17, S18, S19, S2, S21, S6, S8, L1, L14, L15, L16, L17, L18, L19, L20, L23, L28, L29, L3, L31, L32, L33, L4, and L5 were absolutely conserved in the closely related taxa of *H. influenzae* and *H. aegyptius*. The same procedure was applied to MSp from each strain, demonstrating that homologous protein isoforms tend to have similar masses and intensities (Fig. 1). It also became clear that ribosomal proteins S12, S13, S4, S9, L10, L2, L22, L27, and L30 were observed and confirmed to be modified through N-terminal methionine excision and acetylation of N-terminal alanine and serine, consistent with previous studies (38). Furthermore, MALDI-TOF MS analysis identified three additional ribosomal biomarkers: RS10, with *m/z* values of 11,767.6 and 11,636.6; RS12, with *m/z* values of 13,619.2 and 13,662.2; and RL2, with *m/z* values of 30,050.9 and 30,007.9. These biomarkers showed significant differences in their *m/z* values due to various mutations, despite having identical *in silico* calculated *m/z* values, making them highly effective for distinguishing *H. influenzae* and *H. aegyptius*, as shown in Table 2. The ribosomal proteins S13 (*m/z* values of 13,608.9 and 13,134.5), L24 (11,329.2 and 11,129.2), and L27 (9,021.49 and 9,025.4) are also valuable as biomarkers due to their varied calculated masses and possible mutations.

## Proteogenomic prediction of the identified proteins in all sequenced *H. influenzae* and *H. aegyptius* strains

Unique *H. influenzae*-specific proteins were initially identified *in silico* through NCBI protein-versus-translated DNA BLAST analysis against the non-redundant NCBI database. Additionally, proteotyping was performed to identify species-unique peptide sequences derived from LC-MS/MS analysis. This combined approach provides an experimentally validated list of candidate protein markers that can accurately identify organisms at the species level. To enhance the reliability of these biomarkers, it is also important to validate them through detailed comparative genomic analyses.

TABLE 2 Tentative assignment of biomarker peaks obtained by MALDI-TOF MS analyses of *H. influenzae* and *H. aegyptius*, matching with the Hot Protein Database

| Annotated function[a] (amino acid changes relative to H. influenzae CCUG 23945^T) | *Haemophilus influenzae* Calculated sequence mass (m/z) | N-Terminal amino acids | PI | *Haemophilus influenzae* CCUG 23945^T Exptl mass (avg m/z) | CCUG 23969 Exptl mass (avg m/z) | CCUG 26214 Exptl mass (avg m/z) | CCUG 29539 Exptl mass (avg m/z) | CCUG 33775 Exptl mass (avg m/z) | *Haemophilus aegyptius* CCUG 628 Exptl mass (avg m/z) | CCUG 26840 Exptl mass (avg m/z) | CCUG 25716^T Exptl mass (avg m/z) | *Haemophilus aegyptius* Calculated sequence mass (m/z) | N-Terminal amino acids | PI | Maturation |
|---|---|---|---|---|---|---|---|---|---|---|---|---|---|---|---|
| RS2 | 26,133.9 | MA | 6.1 | 26,134.6 | 26,133.8 | 26,134.2 | 26,130 | 26,134.9 | 26,133.92 | 26,133.1 | 26,134.23 | 26,133.94 | MA5 | 6.09 | –Methionine |
| RS2 (+2) | 13,067.5 | | | 13,067.9 | 13,065.8 | 13,067 | 13,065.8 | 13,067.9 | 13,068.4 | 13,068.4 | 13,067.5 | 13,067.5 | | | |
| RS3 (N → S) | 25,722 | MG | 10.32 | | | | | | | 12,940.7 | | 25,880.04 | MG | 10.32 | –Methionine |
| RS3 (+2) | 12,927 | | | 12,927 | 12,928.1 | 12,928.3 | 12,927.3 | 12,927.9 | 12,941.3 | | | 12,940 | | | |
| RS4 (T → A) | 23,437.2 | MA | 10.06 | 23,437.2 | 23,437 | 23,437.8 | 23,437.1 | | | 23,495 | 23,595.6 | 23,495.1 | MA | 10.06 | –Methionine +Acetylation |
| RS4 (+2) | 11,719.1 | | | 11,718.8 | 11,719.5 | 11,719.6 | | | 11,746.2 | 11,747.8 | 11,748.2 | 11,748.1 | | | |
| RS5 (S → A) | 17,370.3 | MS | 10.23 | 17,370.2 | 17,370.5 | 17,372.7 | 17,370.9 | 17,371.3 | 17,385.7 | 17,388.8 | 17,386.8 | 17,386.4 | MS | 10.23 | –Methionine |
| RS6 (+2) | 7177.7 | MR | 5.46 | 7,177.62 | 7,178.43 | | 7,178.32 | 7,177.8 | 7,178.22 | 7,178.22 | 7,178.6 | 7,177.7 | MR | 5.46 | –Methionine |
| RS7 (N → K) | 17,544.3 | MP | 10.19 | 17,544.8 | 17,543.5 | 17,543 | 17,544.2 | 17,544.3 | 17,533.6 | 17,534.7 | 17,533.9 | 17,530.2 | MP | 10.15 | –Methionine |
| RS7 (+2) | 8772.65 | | | | | | | | | | | 8,765.615 | | | |
| RS8 | 13,856.5 | MS | 9.71 | 13,856.6 | 13,855.3 | | 13,855.9 | 13,855.1 | 13,856.7 | 13,875.9 | 13,856.5 | 13,856.5 | MS | 9.71 | –Methionine |
| RS8 (+2) | 6928.75 | | | 6,928.85 | | | 6,928.52 | | | 6,926.38 | 6,926.47 | 6,928.8 | | | |
| RS9 (E → X) | 14,609.2 | MA | 11.09 | | | | | | | | | 14,628 | MA | 10.93 | –Methionine +Acetylation |
| RS9 (+2) | 7305.09 | | | 7,305.22 | 7,306.57 | 7,306.86 | 7,305.23 | 73,07.47 | 7,316.61 | 7,316.56 | 7,314.89 | 7,314.5 | | | |
| RS10 | 11,767.6 | MQ | 9.85 | 11,767.2 | 11,767.1 | 11,766.8 | | | 11,736.2 | 11,636.2 | 11,636.8 | 11,636.6 | MQ | 9.85 | –Methionine |
| R S10 (+2) | 5884.3 | | | 5,885.71 | 5,884.54 | 5,885.15 | | | | | | 5,818.8 | | | |
| R S11 (A → R) | 13,854.2 | MA | 11.49 | | 13,854.7 | 13,851.9 | 13,851.9 | 13,868.2 | 13,768 | 13,760.6 | | 13,769.1 | MA | 11.36 | –Methionine |
| R S11 (+2) | 6927.6 | | | 6,926.98 | 6,927.42 | 6,926.59 | | | | | | 6,885.1 | | | |
| RS12 | 13,619.2 | MA | 10.76 | 13,618.9 | 13,618.6 | 13,619.7 | 13,619.3 | 13,618.2 | 13,661.8 | 13,660.4 | 13,661.5 | 13,662.2 | MA | 10.76 | –Methionine +Acetylation |
| RS12 (+2) | 6810.08 | | | 6,808.82 | 6,809.5 | 6,809.5 | 6,810.94 | 6,809.34 | 6,832.42 | 6,831.45 | 6,831.7 | 6,831.6 | | | |
| RS13 | 13,608.9 | MA | 10.74 | 13,608.5 | 13,608.6 | 13,610.2 | 13,610.2 | 13,608.7 | 13,134.6 | 13,134.5 | 13,132 | 13,134.5 | MA | 10.61 | –Methionine +Acetylation |
| RS13 (+2) | 6804.94 | | | 6,808.16 | 6,810.25 | 6,809.5 | 6,810.25 | 6,809.34 | 6,567.11 | 6,565.09 | 6,565.75 | 6,567.8 | | | |
| RS14 | 11,545.7 | MA | 11 | 11,545 | 11,545.9 | 11,545 | 11,546.4 | 11,544.3 | 11,544 | 11,545 | 11,545.6 | 11,545.7 | MA | 11 | –Methionine |
| RS14 (+2) | 5773.34 | | | 5,772.67 | 5,773.82 | 5,773.04 | 5,774.72 | 5,773.39 | 5,774.49 | 5,772.84 | 5,773.44 | 5,773.3 | | | |
| RS15 | 10,065.8 | MS | 10.69 | 10,065.6 | 10,066.1 | 10,065.2 | 10,065.2 | | 10,065.9 | 10,065.4 | 10,064.8 | 10,065.8 | MS | 10.69 | |
| RS15 (+2) | 5033.4 | | | 5,033.64 | 5,034.67 | 5,033.12 | 5,033.12 | | 5,033.85 | 5,033.17 | 5,033.78 | 5,033.4 | | | |
| RS16 | 9179.55 | MV | 11.4 | 9,178.38 | 9,178.41 | 9,179.4 | 9,178.81 | 9,178.28 | | 9,179.57 | 9,179.72 | 9,179.6 | MV | 11.4 | |
| RS16 (+2) | 4590.27 | | | 4,589.14 | 4,590.33 | 4,591.2 | 4,590.7 | 4,590.82 | | 4,589.67 | 4,589.42 | 4,590.3 | | | |

*(Continued on next page)*

TABLE 2 Tentative assignment of biomarker peaks obtained by MALDI-TOF MS analyses of *H. influenzae* and *H. aegyptius*, matching with the Hot Protein Database (*Continued*)

| Annotated function[a] (amino acid changes relative to *H. influenzae* CCUG 23945^T) | *Haemophilus influenzae* Calculated sequence mass (m/z) | N-Terminal amino acids | PI | Presence in — *Haemophilus influenzae* | | | | | Presence in — *Haemophilus aegyptius* | | | *Haemophilus aegyptius* Calculated sequence mass (m/z) | N-Terminal amino acids | PI | Maturation |
|---|---|---|---|---|---|---|---|---|---|---|---|---|---|---|---|
| | | | | CCUG 23945^T Exptl mass (avg m/z) | CCUG 23969 Exptl mass (avg m/z) | CCUG 26214 Exptl mass (avg m/z) | CCUG 29539 Exptl mass (avg m/z) | CCUG 33775 Exptl mass (avg m/z) | CCUG 628 Exptl mass (avg m/z) | CCUG 26840 Exptl mass (avg m/z) | CCUG 25716^T Exptl mass (m/z) | | | | |
| RS17 (+2) | 4830.24 | MT | 9.94 | 4,829.91 | 4,830.16 | 4,829.4 | 4,829.37 | 4,829.16 | | | | 4,830.24 | MT | 9.94 | −Methionine |
| RS18 | 8812.38 | MA | 10.43 | 8,813.53 | 8,812.86 | 8,810.29 | 8,819.62 | 8,813.74 | 8,813.92 | | | 8,812.4 | MA | 10.43 | −Methionine |
| RS18 (+2) | 4406.69 | | | 4406.86 | | 4,406.62 | | 4407.41 | | | | 4,406.7 | | | |
| RS19 | 10,129.1 | MP | 10.29 | 10,129 | 10,129 | 10,129.5 | 10,130.6 | 10,129.5 | 10,128.1 | 10,129.8 | 10,129 | 10,129.1 | MP | 10.29 | −Methionine |
| RS19 (+2) | 5,065.07 | | | 5,065.58 | 5,065.6 | 5,064.67 | 5,066.18 | | 5,065.54 | 5,064.73 | 5,064.91 | 5,065.1 | | | |
| RS20 (V→A) | 9,508.33 | MA | 10.77 | 9,508.3 | 9,507.74 | 9,508.96 | 9,507.79 | 9,507.95 | 9,506.17 | 9,505.23 | 9,575.94 | 9,508.33 | MA | 10.77 | −Methionine |
| RS20 (+2) | 4,754.66 | | | 4,754.57 | 4,754.56 | 4,754.04 | 4,756.59 | 4,755.15 | 4,755.63 | 4,754.83 | | 4,754.66 | | | |
| RS21 | 8,331.77 | MP | 10.92 | 8,331.74 | 8,331.71 | 8,331.66 | 8,331.26 | 8,333.33 | 8,329.95 | 8,328.88 | 8,329.19 | 8,331.8 | MP | 10.92 | −Methionine |
| RS21 (+2) | 4,166.39 | | | 4,166.42 | 4,167.59 | 4,166.77 | 4,167.13 | 4,169.21 | 4,165.37 | 4,165.59 | 4,165.95 | 4,166.4 | | | |
| RL1 | 23,976.9 | MA | 9.67 | 23,976.9 | 23,976.3 | 23,973.9 | | 23,975.2 | 23,976.3 | 23,978.6 | 23,976.5 | 23,976.9 | MA | 9.67 | −Methionine |
| RL1 (+2) | 12,054.5 | | | 12,054 | 12,054.8 | 12,053.2 | 12,060 | 12,053 | 12,057.8 | 12,056.1 | 12,056.6 | 12,054.5 | | | |
| RL2 | 30,050.9 | MA | 10.79 | 30,050.5 | 30,054.6 | 30,050.1 | 30,051.4 | 30,053.7 | 30,007.8 | 30,007.3 | 30,007.9 | 30,007.9 | MA | 10.79 | |
| RL3 | 23,407.6 | MI | 9.85 | | | | | | | | | 22,407.6 | MI | 9.85 | |
| RL3 (+2) | 11,117.3 | | | 11,117.3 | 11,118.5 | | 11,117.3 | 11,117.8 | 11,116.8 | 11,117.3 | 11,117.2 | 11,117.3 | | | |
| RL4 | 21,955.3 | ME | 9.6 | 21,957.5 | 21,950.9 | 21,956.7 | 21,954.7 | 21,955.5 | | 21,955.5 | 21,955.4 | 21,955.4 | ME | 9.6 | |
| RL4 (+2) | 10,978.2 | | | | 10,970 | | | 10,978.2 | | 10,978.8 | 10,977.7 | 10,978.2 | | | |
| RL5 | 20,354.6 | MA | 9.68 | 20,354.8 | 20,355.3 | 20,354.3 | 20,354.8 | 20,354.4 | 20,359.6 | 20,352.5 | 20,354.3 | 20,354.6 | MA | 9.68 | +Acetylation |
| RL5 (+2) | 10,177.8 | | | 10,177.7 | 10,175.4 | 10,175.5 | 10,176.9 | 10,177.3 | 10,175.2 | 10,176.4 | 10,178.8 | 10,177.8 | | | |
| RL6 (V→G) | 18,948.1 | MS | 9.68 | 18,948.3 | | | | 18,947.3 | 18,990.7 | 18,990.1 | 18,990.5 | 18,990.2 | MS | 9.68 | −Methionine |
| RL6 (+2) | 9,474.53 | | | 9,473.73 | | | | | 9,595.17 | 9,595.23 | 9595.64 | 9495.6 | | | |
| RL7/L12 (A→−) | 12,333.2 | MS | 4.67 | 12,331.8 | 12,333.4 | 12,333 | 12,333 | 12,333.8 | 12,536.5 | 12,536.7 | 12,535.8 | 12,535.3 | MS | 4.67 | −Methionine |
| RL7/L12 (+2) | 6,167.11 | | | 6,167.65 | 6,167.78 | | 6,167.08 | 6,170.71 | 6,268.26 | 6,268.33 | 6,268.79 | 6,268.2 | | | |
| RL9 (T→A) | 15,636.9 | MQ | 6.49 | 15,635.2 | 15,634.1 | 15,636.2 | 15,646 | 15,638.7 | 15,668.2 | 15,667.4 | 15,667.8 | 15,667 | MQ | 6.42 | |
| RL9 (+2) | 7,818.97 | | | 7,817.76 | 7,818.88 | 7,820.65 | 7,819.57 | 7,820.73 | 7,830.19 | 7,834.12 | 7,834.51 | 7,834 | | | |
| RL10 (G→D, A→R & V→A) | 17,545.3 | MA | 8.71 | 17,545.8 | 17,545.5 | 17,543 | 17,546.2 | 17,544.3 | 17,543.8 | 17,561.3 | 17,561.2 | 17,561.4 | MA | 8.68 | −Methionine +Acetylation |
| RL10 (+2) | 8,773.14 | | | | | | | | | 8,788.32 | 8,788.81 | 8,781.2 | | | |
| RL11 (S→T) | 14,904.5 | MA | 9.82 | 14,904.6 | 14,903 | 14,904.7 | 14,904 | 14,904 | 14,890.1 | 14,890.8 | 14,890.2 | 14,890.5 | MA | 9.81 | |
| RL13 (A→V and G→A) | 16,021.6 | MK | 9.83 | 16,000.5 | 15,993.5 | 16,001.8 | 16,003.9 | 16,003.2 | 15,997.1 | 16,000.8 | 15,999.7 | 16,020.7 | MK | 9.83 | +Acetylation |
| RL14 | 13,502 | MI | 10.14 | 13,502.7 | 13,502.5 | 13,502.9 | 13,502.5 | 13,502.8 | 13,502.3 | 13,502.9 | 13,502.6 | 13,502 | | | −Methionine |
| RL14 (+2) | 6,685.98 | MI | 10.14 | 6,685.34 | | | | | | | | | | | |

*(Continued on next page)*

TABLE 2 Tentative assignment of biomarker peaks obtained by MALDI-TOF MS analyses of *H. influenzae* and *H. aegyptius*, matching with the Hot Protein Database (Continued)

| Annotated function[a] (amino acid changes relative to *H. influenzae* CCUG 23945^T) | *Haemophilus influenzae* Calculated sequence mass (m/z) | N-Terminal amino acids | PI | Presence in — *Haemophilus influenzae* CCUG 23945^T Exptl mass (avg m/z) | CCUG 23969 Exptl mass (avg m/z) | CCUG 26214 Exptl mass (avg m/z) | CCUG 29539 Exptl mass (avg m/z) | CCUG 33775 Exptl mass (avg m/z) | *Haemophilus aegyptius* CCUG 628 Exptl mass (avg m/z) | CCUG 26840 Exptl mass (avg m/z) | CCUG 25716^T Exptl mass (avg m/z) | *Haemophilus aegyptius* Calculated sequence mass (m/z) | N-Terminal amino acids | PI | Maturation |
|---|---|---|---|---|---|---|---|---|---|---|---|---|---|---|---|
| RL15 | 15,116.5 | MR | 10.86 | 15,115.8 | 15,115.5 | 15,116.1 | 15,119.1 | 15,117.2 | 15,115.6 | 15,114.6 | 15,113.3 | 15,116.5 | MR | 10.86 | +Acetylation |
| RL16 | 15,102 | ML | 10.9 | 15,101.4 | 15,101.9 | 15,101.9 | 15,101.4 | 15,103.9 | 15,101.6 | 15,103.7 | 15,102.1 | 15,102 | ML | 10.9 | −Methionine |
| RL16 (+2) | 7,551.52 | | | | | | | | | | | 7,551.52 | | | |
| RL17 | 14,473.7 | MR | 10.85 | 14,474.2 | 14,473.3 | 14,474.8 | 14,473 | 14,473.4 | 14,473 | 14,473 | 14,474.3 | 14,473.7 | MR | 10.85 | |
| | 7,237.35 | | | | 7,236.17 | 7,237.39 | | | | | | 7,237.35 | | | |
| RL18 | 12,768.7 | MD | 10.94 | 12,768 | 12,765.9 | 12,768.5 | 12,767.5 | 12,768.9 | 12,766.8 | 12,766.5 | 12,767.7 | 12,768.7 | MD | 10.94 | |
| RL18 (+2) | 6,384.85 | | | 6,382.26 | 6,384.15 | | 6,384.27 | 6,384.24 | 6,382.37 | | | 6,384.9 | | | |
| RL19 | 13,114.2 | MS | 11 | 13,114.6 | 13,114 | 13,117.3 | 13,114.9 | 13,114.1 | 13,114.1 | 13,115.8 | 13,114 | 13,114.2 | MS | 11 | +Acetylation |
| RL19 (+2) | 6,557.6 | | | 6,550.36 | 6,550.48 | 6,550.73 | 6,550.58 | 6,550.6 | 6,554.11 | 6,550.09 | 6,550.75 | 6,557.6 | | | −Methionine |
| RL20 | 13,210.8 | MA | 11.47 | 13,211.4 | 13,210.7 | | 13,210 | | 13,210.6 | 13,211.9 | 13,210 | 13,210.8 | MA | 11.47 | −Methionine |
| RL20 (+2) | 6,605.89 | | | 6,605.72 | 6,604.63 | 6,610.75 | 6,604.37 | 6,605.62 | | 6,605.14 | 6,605.63 | 6,605.9 | | | |
| RL21 (+2) (A → E and X → V) | 11,252.4 | MY | 10.22 | 11,252.9 | 11,254.7 | 11,252.3 | 11,256.8 | 11,252.4 | 11,297.1 | 11,297.5 | 11,297.5 | 11,298.2 | MY | 10.06 | −Methionine |
| RL22 | 12,061.2 | ME | 10.23 | 12,061.6 | 12,062.2 | 12,060.2 | 12,060 | 12,060.1 | 12,061.8 | 12,061.1 | 12,061.6 | 12,061.2 | ME | 10.23 | −Methionine +Acetylation |
| RL22 (+2) | 6,031.11 | | | 6,030.46 | 6,030.77 | | 6,030.23 | 6,031.6 | 6,031.75 | 6,030.69 | 6,030.99 | 6,031.1 | | | |
| RL23 | 10,767.5 | MS | 9.85 | 10,765.2 | 10,768.2 | | 10,768.8 | 10,765.2 | 10,767.4 | 10,766 | 10,767.2 | 10,767.5 | | | −Methionine |
| RL23 (+2) | 5,384.26 | | | 5,383.05 | 5,384.15 | 5,384.36 | 5,385.84 | 5,382.29 | 5,384.77 | | | | | | |
| RL24 (A → P) | 11,329.2 | MP | 10.15 | 11,332.5 | 11,132.5 | 11,332 | 11,331.6 | 11,331.1 | 11,129.3 | 11,129.9 | 11,129.1 | 11,129.2 | MA | 10.15 | −Methionine +Acetylation |
| RL24 (+2) | 5,665.12 | | | | | | | 5,566.33 | | 5,564.91 | 5,564.74 | 5,565.1 | | | |
| RL25 (→) | 10,441.1 | MA | 9.69 | 10,441.6 | 10,441.6 | 10,439.6 | 10,440.6 | 10,440.6 | 10,505.1 | 10,508.9 | 10,508.9 | 10,508.2 | MA | 9.45 | −Methionine |
| RL25 (+2) | 5,221.06 | | | 5,221.2 | 5,221.3 | 5,220.8 | 5,220 | 5,220.4 | 5,252.3 | 5,254.49 | | 5,254.6 | | | |
| RL27 (S → N and S → V) | 9,021.49 | MA | 10.58 | 9,003.93 | 9,004.54 | 9,005.5 | 9,007.65 | 9,006.13 | 9,023.79 | 9,023.62 | 9,024.25 | 9,025.4 | MA | 10.58 | −Methionine +Acetylation |
| RL27 (+2) | 4,511.24 | | | 4,504.66 | 4,505.73 | 4,505.2 | 4,506.02 | 4,505.15 | 4,513.05 | 4,513.16 | 4,514.27 | 4,513.2 | | | |
| RL28 | 8,854.5 | MS | 11.71 | 8,851.08 | 8,850.88 | 8,851.04 | 8,853.62 | 8,851.74 | 8,852.41 | | | 8,854.5 | MS | 11.71 | −Methionine |
| RL28 (+2) | 4,427.75 | | | 4,427.86 | 4,429.14 | 4,428.25 | 4,429.04 | 4,427.73 | 4,428.02 | 4,427.74 | | 4,427.8 | | | |
| RL29 | 7,179.26 | MK | 9.69 | 7,178.12 | 7,178.35 | 7,178.43 | 7,180.32 | 7,178.8 | 7,179.22 | 7,178.22 | 7,178.6 | 7,179.3 | MK | 9.69 | |
| RL29 (+2) | 3,590.13 | | | 3,590.08 | 3,591.07 | 3,590.36 | 3,589.65 | 3,589.89 | 3,589.9 | 3,589.24 | 3,589.39 | 3,590.1 | | | |
| RL30 (M → I) | 6,572.93 | MA | 10.61 | 6,571.09 | 6,572.48 | 6,571.73 | 6,571.58 | 6,571.6 | 6,550.11 | 6,550.09 | 6,550.75 | 6,548 | MA | 10.61 | −Methionine +Acetylation |

**TABLE 2** Tentative assignment of biomarker peaks obtained by MALDI-TOF MS analyses of *H. influenzae* and *H. aegyptius*, matching with the Hot Protein Database (*Continued*)

| Annotated function[a] (amino acid changes relative to H. influenzae CCUG 23945^T) | Haemophilus influenzae Calculated sequence mass (m/z) | N-Terminal amino acids | PI | Presence in | | | | | | | | Haemophilus aegyptius Calculated sequence mass (m/z) | N-Terminal amino acids | PI | Maturation |
|---|---|---|---|---|---|---|---|---|---|---|---|---|---|---|---|
| | | | | Haemophilus influenzae | | | | | Haemophilus aegyptius | | | | | | |
| | | | | CCUG 23945^T Exptl mass (avg m/z) | CCUG 23969 Exptl mass (avg m/z) | CCUG 26214 Exptl mass (avg m/z) | CCUG 29539 Exptl mass (avg m/z) | CCUG 33775 Exptl mass (avg m/z) | CCUG 628 Exptl mass (avg m/z) | CCUG 26840 Exptl mass (avg m/z) | CCUG 25716^T Exptl mass (avg m/z) | | | | |
| RL30 (+2) | 3,286.97 | | | 3,275.15 | 3,274.28 | 3,274.43 | 3,274.35 | 3,273.99 | 3,274.04 | 3,273.62 | 3,273.45 | 3,274.5 | | | |
| RL31 | 7,833.99 | MK | 9.46 | 7,833.88 | 7,833.88 | 7,830.65 | 7,830.73 | 7,829.57 | 7,830.19 | 7,829.12 | 7,829.51 | 7,834 | MK | 9.46 | |
| RL31 (+2) | 3,917.49 | | | 3,916.22 | 3,917.33 | 3,917 | 3,916.87 | 3,916.28 | 3,916.04 | 3,915.49 | 3,915.86 | 3,917.5 | | | |
| RL32 | 6,262.21 | MA | 10.97 | 6,260.29 | 6,260.77 | 6,260.48 | 6,262.98 | 6,260.9 | 6,261.26 | 6,260.33 | 6,260.79 | 6,262.2 | MA | 10.97 | −Methionine |
| RL32 (+2) | 3,131.61 | | | 3,131.84 | 3,131.86 | | 3,132.53 | 3,131.33 | 3,131.47 | 3,132.04 | 3,132.09 | 3,131.6 | | | |
| RL33 | 6,432.77 | MA | 10.06 | 6,431.9 | 6,432.45 | 6,432.28 | 6,432.87 | 6,432.73 | 6,433.12 | 6,432.9 | 6,432.44 | 6,432.8 | MA | 10.06 | −Methionine |
| RL33 (+2) | 3,216.89 | | | 3,216.26 | | 3,216.49 | 3,217.36 | 3,216.82 | 3,216.97 | 3,216 | 3,216.37 | 3,216.9 | | | |
| DNA-binding protein HU (D → N and E → K) | 9,421.84 | MN | 9.4 | 9,420.14 | 9,421.84 | 9,420.8 | 9,421 | 9,421.16 | 9,420.61 | 9,419.46 | 9,419.66 | 9,419.9 | MN | 9.4 | |
| DNA-binding protein HU (+2) | 4,711.42 | | | 4,711.56 | 4,711.42 | 4,712.22 | 4,712.82 | 4,712.13 | 4,710.6 | 4,710.96 | 4,710.06 | 4,710.5 | | | |
| Ribosomal RNA large subunit methyltransferase H | 18,027.1 | MN | 6.39 | 18,033.1 | 18,033.4 | 18,033.1 | 18,033.7 | 18,036.9 | 18,035.8 | 18,067.9 | 18,067.9 | 18,072.1 | MN | 6.39 | |
| Ribosomal RNA large subunit methyltransferase G | 15,667.1 | MT | 5.57 | | 15,659.9 | | 15,661.1 | 15,664.3 | 15,668.2 | 15,660.4 | 15,665.4 | 15,666.6 | MT | 5.81 | −Methionine |
| Ribosomal RNA large subunit methyltransferase J | 11,546 | MK | 8.56 | 11,541 | 11,542.9 | 11,541 | 11,543.4 | 11,542.3 | 11,543 | 11,541 | 11,541.6 | 11,540.4 | MK | 7.81 | |
| Ribosomal RNA large subunit methyltransferase | 16,109.5 | ML | 8.37 | 16,109.2 | 16,110.3 | 16,120.7 | 16,116.7 | 16,111.4 | 16,187.3 | 16,187.4 | 16,185.4 | 16,186.7 | ML | 9.09 | −Methionine |
| Ribosomal RNA large subunit methyltransferase A | 16,198.3 | MN | 7.21 | 16,193.7 | 16,195 | 16,185.2 | 16,192.7 | 16,199.9 | 16,306.1 | 16,303 | 16,306.6 | 16,299.3 | MN | | −Methionine |

**TABLE 2** Tentative assignment of biomarker peaks obtained by MALDI-TOF MS analyses of *H. influenzae* and *H. aegyptius*, matching with the Hot Protein Database (*Continued*)

| Annotated function[a] (amino acid changes relative to *H. influenzae* CCUG 23945^T) | *Haemophilus influenzae* Calculated sequence mass (m/z) | N-Terminal amino acids | PI | Presence in — *Haemophilus influenzae* CCUG 23945^T Exptl mass (avg m/z) | CCUG 23969 Exptl mass (avg m/z) | CCUG 26214 Exptl mass (avg m/z) | CCUG 29539 Exptl mass (avg m/z) | CCUG 33775 Exptl mass (avg m/z) | *Haemophilus aegyptius* CCUG 628 Exptl mass (avg m/z) | CCUG 26840 Exptl mass (avg m/z) | CCUG 25716^T m/z | *Haemophilus aegyptius* Calculated sequence mass (m/z) | N-Terminal amino acids | PI | Maturation |
|---|---|---|---|---|---|---|---|---|---|---|---|---|---|---|---|
| Ribosomal RNA large subunit methyltransferase E | 23,452 | MR | 6.67 | 23,448.6 | 23,448.4 | 23,453.8 | 23,444.5 | | | 23,385.6 | | 23,398 | MR | 6.4 | –Methionine |
| Ribosomal RNA large subunit methyltransferase E (+2) | 11,726.5 | | | 11,719.5 | | | 11,721.6 | | | 11,691.8 | 11,695.3 | 11,699.5 | | | |
| [Ribosomal protein S18]-alanine N-acetyltransferase | 16,615.4 | MS | 5.88 | 16,780.8 | | 16,796.7 | 16,615.2 | 16,615 | 16,590.4 | 16,591.4 | 16,591 | 16,591.4 | MS | 5.88 | –Methionine |
| [Ribosomal protein S18]-alanine N-acetyltransferase (+2) | 8,308.18 | | | 8,383.33 | 8,386.59 | 8,385.43 | 8,300.48 | 8,315.2 | 8,302.51 | 8,301.27 | 8,301.75 | 8,296.2 | | | |

[a]Only the putatively protein peaks that have been assigned are displayed. Table S2 displays a complete peak list, including unidentified biomarker ions.

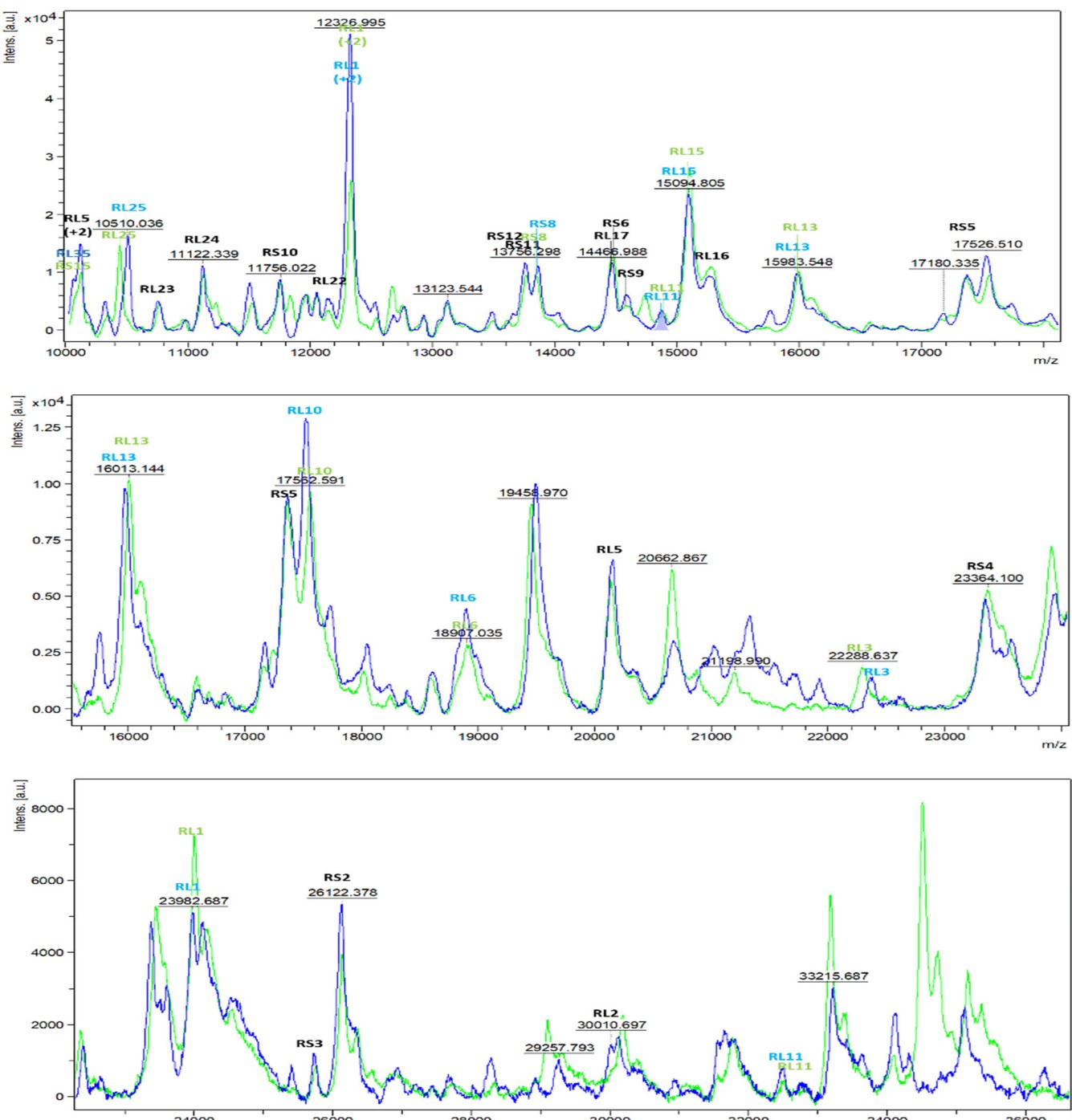

**FIG 1** Representative MALDI mass spectra for strains of *H. influenzae* CCUG 23945[T] (green) and *H. aegyptius* CCUG 25716[T] (blue), based on variations in protein biomarkers.

## Comprehensive protein analysis and phylogenetic validation of biomarkers

Advances in next-generation sequencing (NGS) have enabled the sequencing of whole genomes at a low cost with high throughput (39, 40), resulting in an exponential growth in the number of genome sequences over the past decade. However, this surge in genomic data has highlighted issues with the accuracy and completeness of reference databases, including errors in genome annotations, variations in sequence quality, and misidentifications, underscoring the need for minimal standards in the use of genome

data (41). To address these challenges and systematically evaluate species boundaries, average nucleotide identity (ANI) values of the 892 selected genomes were calculated using pyANI 0.2.11, providing genomic similarity profiles for each strain; ANI values of ≥94%–96% were considered indicative of strains belonging to the same species, whereas values below this range were regarded as representing different species, according to the established species delineation criteria (41, 42). ANI values ranged from 95% to 97% between the genomes of the two species, indicating a close but distinct relationship. The genomes were clustered based on their similarity to the reference strains according to ANI analysis. From these clusters, a representative subset of 93 genomes closely matching the type strains was chosen for further analysis. This subset included 76 strains associated with *H. influenzae* NCTC 8143$^T$ (=CCUG 23945$^T$) and 17 strains closely related to *H. aegyptius* NCTC 8502$^T$ (=CCUG 25716$^T$). These representative genomes were then used for the genomic validation of protein candidate biomarkers. This approach allowed the identification of potential variations among the 39 proteins between *H. influenzae* and *H. aegyptius*. Additionally, it enabled *in silico* identification of species-specific tryptic peptides that distinguish the two species, which could then be compared to the peptides identified by the LC-MS/MS proteotyping method.

To ensure a thorough analysis, protein sequences from each genome were individually extracted, allowing for an in-depth investigation of the amino acid sequence variation associated with the identified biomarkers. This confirmed the presence of significant variations and enabled the identification of potential differences among 39 proteins between *H. influenzae* and *H. aegyptius*. The comparative analysis of protein sequences revealed notable amino acid variations across a specific set of proteins, each providing unique and valuable information for distinguishing between the two species. The presence or absence of certain amino acid changes within these proteins serves as a reliable basis for species differentiation.

Of the 39 candidate protein biomarkers identified from the limited set of strains analyzed by MALDI-TOF MS, based on the comparative analysis, 31 proteins were selected for inclusion in the study due to their distinct amino acid variations, which likely contribute to the species-specific traits of *H. influenzae* and *H. aegyptius*. Conversely, eight proteins were excluded from the analysis because they lacked distinguishing amino acid changes or had a limited impact on species differentiation. This strategic selection of proteins ensures a focused approach to identify key biomarkers that distinguish *H. influenzae* from *H. aegyptius*, thereby improving the accuracy and reliability of species identification (Table S5). Among the included proteins, several tryptic peptides (both *in silico* predicted and experimentally verified) showed notable differences between the two species, further supporting their potential as peptide biomarkers (18, 19, 22), and the differences in amino acid sequences also allowed the use of the protein itself as a biomarker, with detectable molecular mass shifts in the MALDI-TOF-MS spectra.

To further emphasize how the selected proteins could serve as biomarkers for species differentiation, a phylogenetic analysis was conducted. Protein sequences were aligned using ClustalW, and an MLSA of concatenated proteins was constructed. The Neighbor-Joining method, with a bootstrap value of 1,000, was used to generate a phylogenetic tree (Fig. 2). This analysis confirmed the suitability of the chosen biomarkers, revealing species-specific variations that supported the differentiation between *H. influenzae* and *H. aegyptius*. During the analysis, some protein sequences were incomplete or missing in certain genomes based on BLASTp results. These proteins, although absent in some genomes, were not excluded from the Neighbor-Joining tree to provide a comprehensive overview, with gaps in the sequences indicated by "–."

Additionally, a second phylogenetic analysis was conducted, this time excluding the proteins recommended for removal based on the initial analysis (Fig. 2). Both analyses, with and without the non-variable proteins, consistently showed clear clustering of *H. aegyptius* and *H. influenzae*, emphasizing the distinct evolutionary relationships between the two species. This further affirms the role of the selected proteins as reliable biomarkers for differentiating these closely related species.

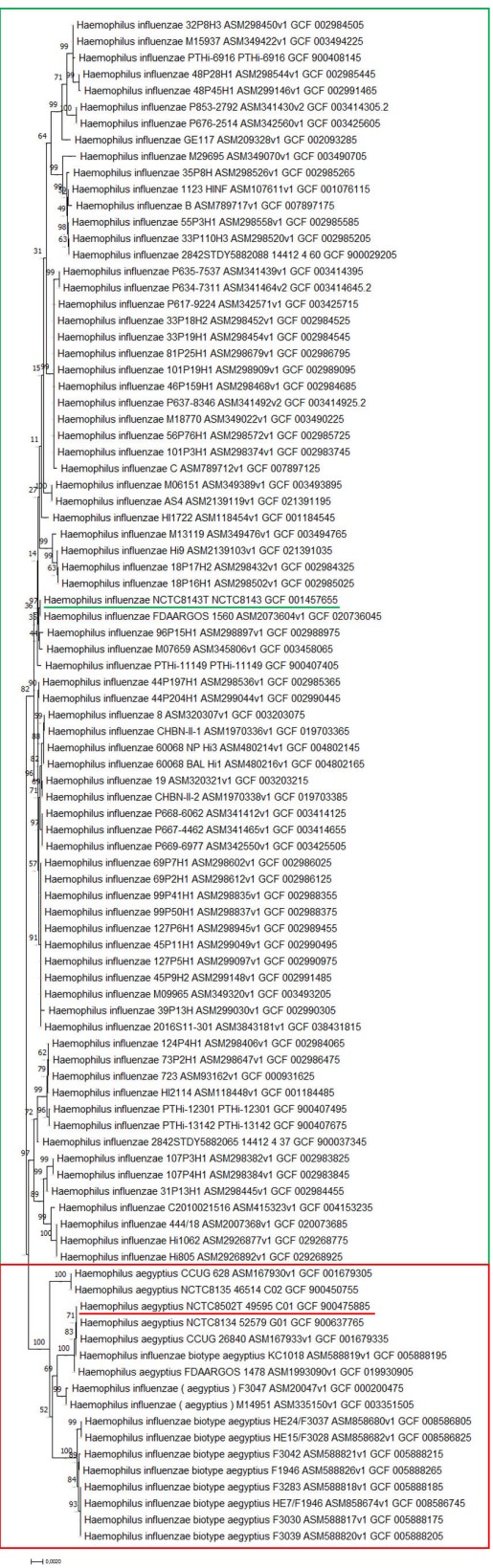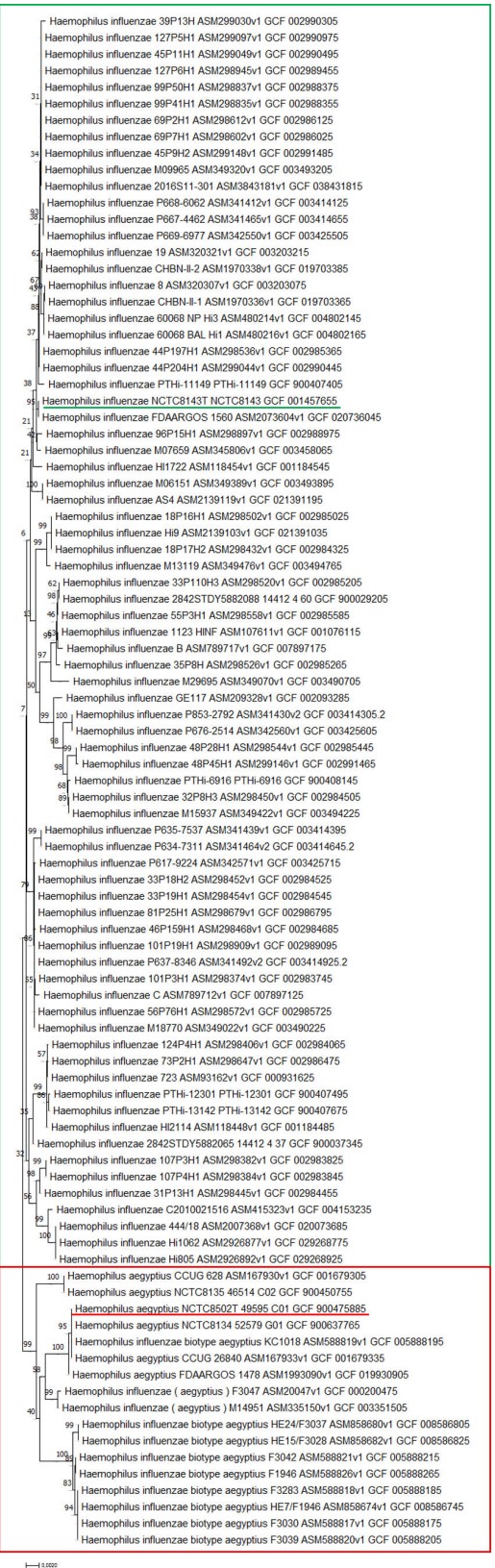

**FIG 2** MLSA resultant phylogenetic trees inferred using the neighbor-joining method for 93 genomes of *H. influenzae* and *H. aegyptius*. (**Left**) Tree using 39 biomarker proteins. (**Right**) Tree using 31 reliable biomarker proteins, with excluded proteins from the initial analysis omitted. The amino acid substitution model employed was the Poisson model, and the bootstrap consensus was inferred from 10,000 replicates. The green square indicates *H. influenzae*, with the type strain highlighted, while the defined group of *H. aegyptius* is shown in red.

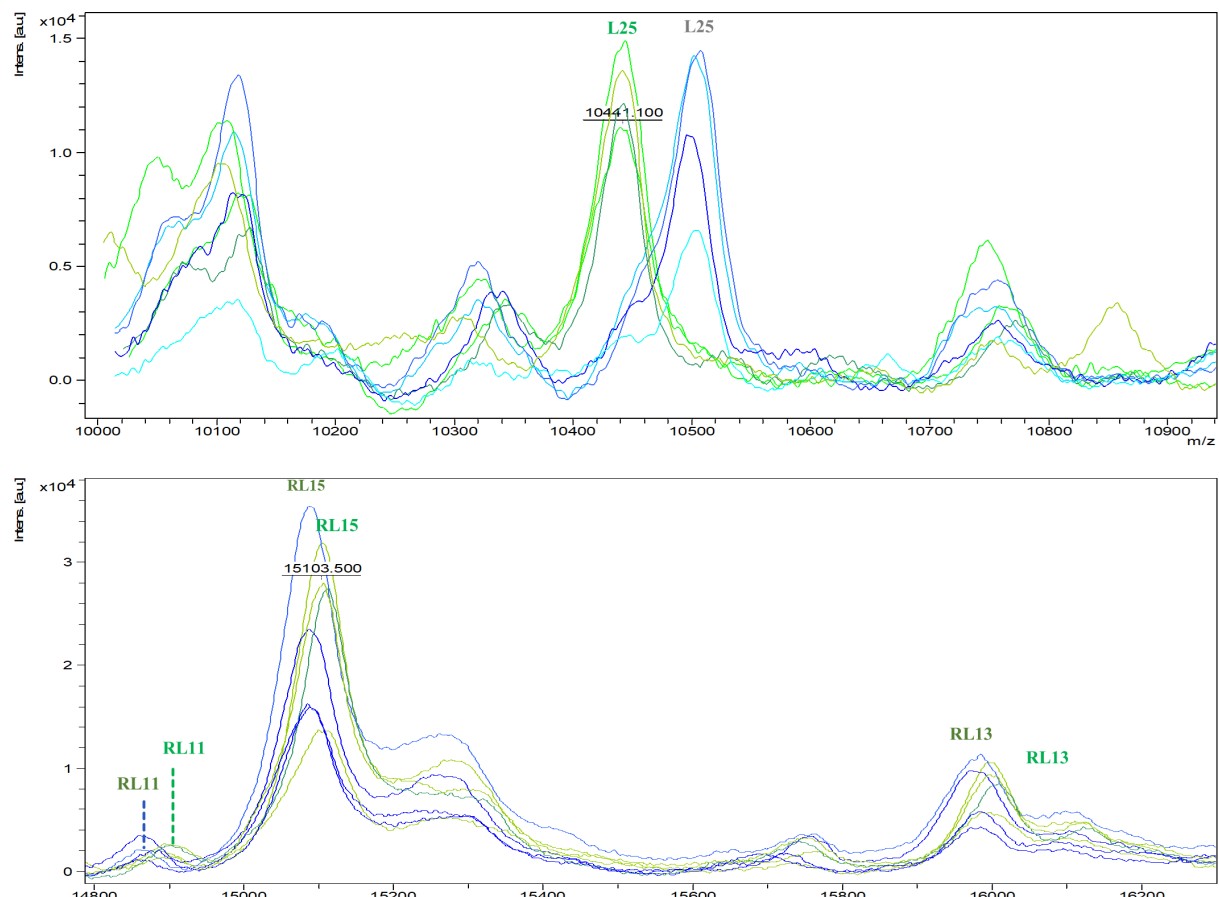

**FIG 3** Subregion of MALDI mass spectra for strains of *H. influenzae* vs *H. aegyptius*. Overlay of MALDI mass spectra of *H. influenzae* CCUG 26214. *H. influenzae* CCUG 29539, *H. influenzae* CCUG 23969, and *H. influenzae* CCUG 23945[T] (green). Overlay of MALDI mass spectra of *H. aegyptius* CCUG 26840, *H. aegyptius* CCUG 25716[T], and *H. influenzae* CCUG 628 (blue).

## Biomarker discovery and verification

The final analysis of proteins and their peptides has identified potential biomarkers for distinguishing between the closely related species, *H. influenzae* and *H. aegyptius*, using MALDI-TOF MS. Each identified protein plays a unique role in classification, with specific amino acid differences providing a solid basis for accurate species identification. Notably, several proteins showed significant signal peak intensities during MALDI-TOF MS analysis, responding effectively to both ion charge states (1+ or 2+). Key proteins include ribosomal RNA large subunit methyltransferase H (RlmH), which displays amino acid changes (S → F & G → S), and DNA-binding protein HU (HU), which produced strong signals with variations (D → N & E → K). Additional contributions came from ribosomal proteins (RL6, RL7/L12, RL11, RL21, RL25, RL24, RL27, RS3, RS5, and RS7) and various outer membrane protein assembly factors, further enhancing our understanding of the molecular relationships between these species (Fig. 3).

While most of the identified proteins are ribosomal, the tentative classification of non-ribosomal proteins reflects the difficulty of accurately distinguishing these species; relying solely on ribosomal proteins is inadequate. Therefore, a comprehensive approach that includes a wider range of proteins is necessary for effective species classification (Table 3). This analysis emphasizes the importance of using MALDI-TOF MS for reliable biomarker identification and investigating proteomic differences among closely related bacterial species. Such advancements are vital for enhancing diagnostic capabilities in microbiology.

**TABLE 3** Predicted MALDI-TOF *m/z* values for the monocharged biomarkers according to their species and strains

| Species | Strains | RL6 | RL7/L12 | RL11 | RL21 | RL24 | RL25 | RL27 | RlmH | HU | RS5 | RS7 |
|---------|---------|-----|---------|------|------|------|------|------|------|----|----|----|
| *Haemophilus* | CCUG 23945[T] | 18,948.3 | 12,331.8 | 14,904.6 | 11,252.9 | 11,332.5 | 10,441.6 | 9,003.93 | 18,033.1 | 9,420.14 | 17,370.2 | 17,544.8 |
| *influenzae* | CCUG 23,969 | 18,948.2 | 12,333.4 | 14,903 | 11,254.7 | 11,132.5 | 10,441.6 | 9,004.54 | 18,033.4 | 9,421.84 | 17,370.5 | 17,543.5 |
| | CCUG 26,214 | 18,947.1 | 12,333 | 14,904.7 | 11,252.3 | 11,332 | 10,439.6 | 9,005.5 | 18,033.7 | 9,420.8 | 17,372.7 | 17,543 |
| | CCUG 29,539 | 18,948.7 | 12,333 | 14,904 | 11,256.8 | 11,331.6 | 10,440.6 | 9,007.65 | 18,036.9 | 9,421 | 17,370.9 | 17,544.2 |
| | CCUG 33,775 | 18,947.3 | 12,333.8 | 14,904 | 11,252.4 | 11,331.1 | 10,440.6 | 9,006.13 | 18,035.8 | 9,421.16 | 17,371.3 | 17,544.3 |
| *Haemophilus* | CCUG 628 | 18,990.7 | 12,536.5 | 14,890.1 | 11,297.1 | 11,129.3 | 10,505.1 | 9,023.79 | 18,067.9 | 9,420.61 | 17,385.7 | 17,533.6 |
| *aegyptius* | CCUG 26,840 | 18,990.1 | 12,536.7 | 14,890.8 | 11,297.5 | 11,129.9 | 10,508.9 | 9,023.62 | 18,067.9 | 9,419.46 | 17,388.8 | 17,534.7 |
| | CCUG 25716[T] | 18,990.5 | 12,535.8 | 14,890.2 | 11,297.5 | 11,129.1 | 10,508.9 | 9024.25 | 18,072.1 | 9419.66 | 17,386.8 | 17,533.9 |

This study shows that high-resolution proteotyping, which combines whole-cell MALDI-TOF MS with nanoLC-MS/MS, allows for reliable differentiation of *Haemophilus influenzae* and *Haemophilus aegyptius*, two clinically important and genetically similar species. By extending the MALDI-TOF MS mass range to *m/z* 40,000 and pairing spectral analysis with detailed genomic and proteomic validation, we identified 31 distinct protein biomarkers that are consistently and uniquely linked to *H. influenzae* and *H. aegyptius*. These differences highlight their potential for accurate species identification. The results from these genomic validations emphasize key distinguishing features, confirming their usefulness as dependable, species-specific biomarkers.

Furthermore, genomic and phylogenetic validations not only confirmed the uniqueness of these proteins but also demonstrated their usefulness through clear evolutionary clustering of the two species. *In silico* predictions and experimental testing of tryptic peptides further supported the specificity of these biomarkers, ensuring their robustness and applicability. Overall, these results reinforce the role of these biomarkers as dependable tools for accurately identifying and distinguishing *H. influenzae* from closely related species.

This combined approach overcomes the limitations of current phenotypic and genomic methods, providing improved taxonomic resolution that is relevant to clinical practice. Importantly, our findings highlight the need for ongoing refinement of reference spectral and genomic databases to ensure accurate pathogen identification in diagnostic microbiology. The proposed methodology and biomarker set establish a foundation for enhancing diagnostic accuracy, especially in differentiating pathogens with similar phenotypes but different pathogenic potentials. Incorporating such targeted proteotyping strategies into clinical workflows may ultimately lead to more effective treatments and better patient outcomes.

## ACKNOWLEDGMENTS

The CCUG and the laboratory staff are acknowledged for providing reference strains and expert characterization and maintenance. The CCUG is supported by the Department of Clinical Microbiology, Sahlgrenska University Hospital, and the Sahlgrenska Academy of the University of Gothenburg, Sweden. The staff of the Bacteriology laboratory of the Department of Clinical Microbiology of Sahlgrenska University Hospital is acknowledged for providing clinical samples and for expert identification analyses. We acknowledge Dr. Miriam Cordovana and Dr. Markus Kostrzewa at Bruker Daltonics for their support, insight, and fruitful discussion. The authors thank the Proteomics Core Facility, University of Gothenburg, co-funded by SciLifeLab and BioMS/Vetenskapsrådet, for expertise in mass spectrometry-based proteomics analyses.

This study was funded by the European Union's Horizon 2020 research and innovation program under the Marie Sklodowska-Curie grant agreement 955626, regional FOUS VGFOUREG-969330, VGFOUREG-994802, VGFOUREG-665141, and ALF ALFGBG-966570.

Conceptualization: H.H., A. Gillberg, E.R.B.M., and R.K.; Methodology: H.H., A. Gillberg, and R.K.; Validation: H.H., E.R.B.M., G.S., and R.K.; Formal analysis: H.H., G.S., and R.K.; Investigation: H.H., G.S., and R.K.; Resources: E.R.B.M. and R.K.; Data Curation: H.H., G.S.,

and R.K.; Writing - Original Draft: H.H., G.S., and R.K.; Writing - Review & Editing: H.H., G.S., A. Ghassempour, E.R.B.M., and R.K.; Visualization: H.H., G.S., and R.K.; Supervision: A. Ghassempour, E.R.B.M., and R.K.; Project administration: E.R.B.M. and R.K.; Funding acquisition: E.R.B.M. and R.K.

## AUTHOR AFFILIATIONS

[1]Department of Infectious Diseases, University of Gothenburg, Institute of Biomedicine, Sahlgrenska Academy, Gothenburg, Sweden

[2]Department of Phytochemistry, Shahid Beheshti University, Medicinal Plants and Drugs Research Institute, Tehran, Iran

[3]Department of Clinical Microbiology, Sahlgrenska University Hospital, Gothenburg, Sweden

[4]Culture Collection University of Gothenburg (CCUG), Sahlgrenska University Hospital and Sahlgrenska Academy, University of Gothenburg, Gothenburg, Sweden

## AUTHOR ORCIDs

Guillem Seguí  http://orcid.org/0000-0001-8739-3289
Edward R. B. Moore  http://orcid.org/0000-0001-7693-924X
Roger Karlsson  http://orcid.org/0000-0002-5919-2639

## FUNDING

| Funder | Grant(s) | Author(s) |
| --- | --- | --- |
| Västra Götalandsregionen | ALFGBG-966570 | Edward R. B. Moore |
| Västra Götalandsregionen | VGFOUREG-665141 | Roger Karlsson |
| Västra Götalandsregionen | VGFOUREG-969330 | Roger Karlsson |
| Västra Götalandsregionen | VGFOUREG-994802 | Roger Karlsson |
| European Commission | 955626 | Edward R. B. Moore |

## AUTHOR CONTRIBUTIONS

Hamideh Hamidi, Conceptualization, Data curation, Formal analysis, Investigation, Methodology, Validation, Visualization | Guillem Seguí, Data curation, Formal analysis, Investigation, Validation, Visualization | Alexandra Gillberg, Conceptualization, Methodology | Alireza Ghassempour, Supervision | Edward R. B. Moore, Conceptualization, Project administration, Resources, Supervision, Validation | Roger Karlsson, Conceptualization, Data curation, Formal analysis, Investigation, Methodology, Project administration, Resources, Supervision, Validation, Visualization

## DATA AVAILABILITY

The mass spectrometry proteomics data have been deposited in the MassIVE partner repository with the data set identifier. Title: Proteotyping-Based Differentiation of *H. influenzae* and *H. aegyptius* and MassIVE ID: MSV000099548.

## ADDITIONAL FILES

The following material is available online.

### Supplemental Material

**Supplemental material (Spectrum03542-25-s0001.docx).** Optimization of analytical procedures; Fig. S1 and S2.
**Table S1 (Spectrum03542-25-s0002.xlsx).** Theoretical masses of ribosomal subunit proteins of *H. influenzae* used for internal recalibration.

**Table S2 (Spectrum03542-25-s0003.xlsx).** MSP describing m/z values, corresponding average intensities, and peak occurrence frequency of the included strains.

**Table S3 (Spectrum03542-25-s0004.xlsx).** LC-MS/MS proteotyping results showing identification of proteins of *H. influenzae* CCUG 23945[T] in the molecular range of 3-40 kDa.

**Table S4 (Spectrum03542-25-s0005.xlsx).** LC-MS/MS proteotyping results showing identification of proteins of *H. aegyptius* CCUG 27516[T] in the molecular mass range of 3-40 kDa.

**Table S5 (Spectrum03542-25-s0006.xlsx).** Biomarker protein variants among sequenced *H. influenzae* and *H. aegyptius* strains.

## Open Peer Review

**PEER REVIEW HISTORY (review-history.pdf).** An accounting of the reviewer comments and feedback.

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
