## [Reviewer comments · Microbiology Spectrum]

Microbiology Spectrum

High-Resolution Differentiation and Identification of the Clinically-Relevant pathogens *Haemophilus influenzae* and *Haemophilus aegyptius*: Combined Whole-Cell MALDI-TOF MS and NanoLC-MS/MS Proteotyping for Defining Protein Biomarkers

Hamideh Hamidi, Guillem Seguí, Alexandra Gillberg, Alireza Ghassempour, Edward Moore, and Roger Karlsson

Corresponding Author(s): Roger Karlsson, Sahlgrenska universitetssjukhuset

Review Timeline:

Submission Date:	November 3, 2025
Editorial Decision:	December 8, 2025
Revision Received:	January 29, 2026
Accepted:	February 5, 2026

Editor: Takeaki Wajima

Reviewer(s): The reviewers have opted to remain anonymous.

Transaction Report:

DOI: <https://doi.org/10.1128/spectrum.03542-25>

Re: Spectrum03542-25 (High-Resolution Differentiation and Identification of the Clinically-Relevant pathogens *Haemophilus influenzae* and *Haemophilus aegyptii*: Combined Whole-Cell MALDI-TOF MS and NanoLC-MS/MS Proteotyping for Defining Protein Biomarkers)

Dear Dr. Roger Karlsson:

Thank you for the privilege of reviewing your work. Below you will find my comments, instructions from the Spectrum editorial office, and the reviewer comments.

Both reviewers recognize the significance and technical quality of your study. However, they have provided detailed comments regarding terminology, methodological clarity, text structure, and the need for additional data or validation. Please revise the manuscript carefully in accordance with these suggestions.

Revision Guidelines

Sincerely,
Takeaki Wajima
Editor
Microbiology Spectrum

Reviewer #1 (Comments for the Author):

The Spectrum 03542-25 manuscript aims to identify protein biomarkers for clinically relevant *Haemophilus* pathogens. This is an

impressive study, integrating data from 892 genomes, whole-cell MALDI-TOF, and nanoLC-MS/MS proteomics. The work is clearly presented and highly convincing. I am confident it will be of significant interest to the clinical diagnostics community, particularly for rapid MALDI-TOF-based pathogen typing. I recommend publication of this sound study. Below are a few suggestions to further improve the presentation of the data.

1. The authors state (line 191) that the isolation window was set to 1.2 Da for tandem mass spectrometry. This is incorrect, as the isolation window applies to mass-to-charge (m/z) values rather than molecular mass. Although this type of mistake is common in the literature, it should be corrected. Even the expression "1.2 m/z window" is technically shorthand, because m/z is a dimensionless ratio rather than a true unit. According to IUPAC recommendations, m/z (italicized, no spaces) should be used to denote mass-to-charge values, the use of the thomson (Th) is no longer recommended. Therefore, the authors should revise the statement to: "The isolation window was set to m/z 1.2."
2. The authors mentioned (line 33, line 153, and line 464) « 40,000 m/z » while it should be « m/z 40,000 »! ... or m/z 3,000 to 40,000.
3. Please review the entire document and ensure that m/z is consistently italicized throughout.
4. I believe Suarez was the first, in 2013, to apply a strategy combining MALDI-TOF and proteomics to identify markers for *Neisseria meningitidis*. This pioneering work may be of interest for readers of the journal.

Reviewer #2 (Comments for the Author):

Please refer to the attached PDF document.

Spectrum03542-25, "*High-Resolution Differentiation and Identification of the Clinically-Relevant pathogens Haemophilus influenzae and Haemophilus aegyptius: Combined Whole-Cell MALDI-TOF MS and NanoLC-MS/MS Proteotyping for Defining Protein Biomarkers*" by Hamidi et al.

In this work, Hamidi and colleagues describe an approach to extending the MALDI-ToF mass spectrometry database with the aim of improving the identification of species from the genus *Haemophilus*, specifically for discriminating between the closely related species of *Haemophilus influenzae* and *Haemophilus aegyptius*.

A variety of analytical techniques were employed to characterize isolates from both *Haemophilus* species, including whole cell MALDI-ToF MS, tandem nanoLC-MS and comparative genomics. The aim was to generate high-quality MALDI-ToF mass spectra that could be used to reliably differentiate between the two species and identify them. High-resolution tandem LC-MS and genomic analyses were then performed to detect discriminatory, or identifying biomarkers and improve the understanding of the molecular basis for identification and discrimination.

Overall, the study was conducted carefully and is principally relevant due to the varying clinical and veterinary significance of the characterized *Haemophilus* species. The data and results provided are interesting and reliable, and the mass spectrometry techniques used are state of the art. I have no doubts about the validity of the presented data and of most of the results. However, I must say that the conclusions are not always supported by the data and that the way of presenting data and results requires major improvements. While some relevant aspects are only described in part, other points are often listed redundantly in the text. The manuscript would benefit from a more streamlined text, improved structure and a clearer focus on the work performed. Collecting additional data seems to be required (see below)

Furthermore, I would have liked the MALDI spectra of *Haemophilus* species mentioned in the manuscripts to be made available for the review process, at least. In this era of open data, it is important that the data acquired in the course of the study should be available for verification of the study results.

Figures and tables: I believe that a detailed list of the microbial isolates examined should be included at the beginning of the manuscript (see Table 1). In my opinion, Table 1 (theoretical masses of *H. influenzae* for internal calibration) is not particularly important in the current version and could therefore be moved to the supplementary information (SI).

My concerns, comments and suggestions regarding important aspects of the investigations are detailed below.

1. My main criticism is that only a very small number of *H. influenzae* and *H. aegyptius* strains were examined (six *H. influenzae* and three *H. aegyptius*). In my opinion, it is not possible to draw robust conclusions from the examination of such a small number of strains. I believe that independent validation of the results would be necessary. At the very least, additional strains of both species (ideally $n > 10$) should be characterized according to a MALDI standard protocol. The recorded spectra would then need to be used to verify whether the discriminatory markers

postulated in the manuscript were found in the independent samples (i.e. independent validation).

2. To determine whether the described biomarkers are specific to the respective species, the study should also include other closely and distantly related species (e.g. *H. parainfluenzae*, *H. quentini* and *H. haemolyticus* ssp. *intermedius*, see [5]). This is the only way to determine whether the biomarkers are truly identifying the species or if they are also found in other species of the genus.
3. Section MALDI-ToF mass spectrometry, page 4 lines 126-135. MALDI-ToF mass spectrometry of bacteria is a widely employed technique with thousands of published studies. Against this background, I find it incomprehensible that new methods of sample preparation and for MALDI measurements were established and optimized instead of using one of the standard methods. While using alternative methods is certainly possible, such methods must be then documented in much greater level of detail. However, this section description contains only a limited amount of information, and I believe that the methodology is not comprehensible to third parties.
4. Page 8, line 153. What is a MALDI Biotyper smart instrument? Bruker does not offer such an instrument. Please correct.
5. Section NanoLC-MS/MS Analysis of Proteotyping Samples Page 9: (As a comment only.) These measurements were performed in DDA mode. DIA mode acquisition schemes offer much more complete proteomic information, so it is unclear why this somewhat outdated methodology was used.
6. It is not clear to me how and why spectra were transferred from the Bruker MBT ecosystem to the SARAMIS software from bioMérieux. Firstly, it would be interesting to know how exactly the spectral data recorded on a Bruker Daltonics mass spectrometer was imported into the SARAMIS software. I also find it difficult to understand why the Bruker software (MBT) was not used for the analysis. The logical approach for this study would be to first use MBT and the RUO database to identify the recorded data, and then expand the existing databases to include MSPs from *Haemophilus influenzae* and *Haemophilus aegyptius*. Subsequently, identification of additional, previously unused strains should be based on the expanded, customer-specific database. It is unclear why the authors did not take this approach, which has already been used in many other studies (see also my previous comments).
7. The methodological approach of combining MALDI-ToF MS and tandem LC-MS to increase the understanding of the molecular basis, as well as the accuracy of identification and differentiation, is adequate but not particularly novel. Other authors have previously used this and similar approaches in earlier studies to improve understanding of MALDI-based diagnostics, e.g. for biomarker discovery and validation. Therefore, I think it would be fair to cite studies such as Fagerquist (2016) Expert Rev Proteomics [1], Lasch et al. (2016) TRAC [2], and Dieckmann et al. (2008) AEM [3], perhaps also ref. [4]. To avoid excessive citation, some redundant citations in the 'Introduction' section could be omitted (lines 62–63, page 4, citations #1–#6).
8. There is some overlap in the statements in the two sections, 'Abstract' and 'Importance'. Perhaps the authors could take another critical look at both paragraphs and try to reduce the redundancy of their statements.
9. Pages 12 and 13, section 'Results and Discussion': This long section reads like a second introduction. I suggest significantly shortening this section and moving it to the introduction, as it

contains redundant information. The results section should start with a description of the results.

10. Page 18, In the strict sense, ANI values larger than 95% (sometimes 96% are stated) do not support the definition of *H. influenzae* and *H. aegyptius* as separate species. Genomes sharing an ANI value above this threshold are generally considered to belong to the same species, while those with values below 90% are typically considered different species. This could be mentioned in somewhere in the manuscript.
11. Page 20, line 440. Please complete this sentence.

[1] <http://www.ncbi.nlm.nih.gov/pubmed/27838927>

[2] <http://dx.doi.org/10.1016/j.trac.2016.04.013>

[3] <http://www.ncbi.nlm.nih.gov/pubmed/18952875>

[4] <http://www.ncbi.nlm.nih.gov/pubmed/23916798>

[5] <https://www.ncbi.nlm.nih.gov/pubmed/35139905>

Reviewer #1 (Comments for the Author):

The Spectrum 03542-25 manuscript aims to identify protein biomarkers for clinically relevant *Haemophilus* pathogens. This is an impressive study, integrating data from 892 genomes, whole-cell MALDI-TOF, and nanoLC-MS/MS proteomics. The work is clearly presented and highly convincing. I am confident it will be of significant interest to the clinical diagnostics community, particularly for rapid MALDI-TOF-based pathogen typing. I recommend publication of this sound study. Below are a few suggestions to further improve the presentation of the data.

1. The authors state (line 191) that the isolation window was set to 1.2 Da for tandem mass spectrometry. This is incorrect, as the isolation window applies to mass-to-charge (m/z) values rather than molecular mass. Although this type of mistake is common in the literature, it should be corrected. Even the expression "1.2 m/z window" is technically shorthand, because m/z is a dimensionless ratio rather than a true unit. According to IUPAC recommendations, m/z (italicized, no spaces) should be used to denote mass-to-charge values, the use of the thomson (Th) is no longer recommended. Therefore, the authors should revise the statement to: "The isolation window was set to m/z 1.2."

Author response: *We agree with the reviewer and have changed the sentence accordingly.*

2. The authors mentioned (line 33, line 153, and line 464) « 40,000 m/z » while it should be « m/z 40,000 »! ... or m/z 3,000 to 40,000.

Author response: *We agree. We have changed throughout the text, for example, the lines 33, 152, 284, 292, 308, 322, 334, 462.*

3. Please review the entire document and ensure that m/z is consistently italicized throughout.

Author response: *All m/z have been consistently italicized.*

4. I believe Suarez was the first, in 2013, to apply a strategy combining MALDI-TOF and proteomics to identify markers for *Neisseria meningitidis*. This pioneering work may be of interest for readers of the journal.

Author response: *We agree with the Reviewer and we have added this reference to the introduction at page 6 line 126-130.*

“To improve MALDI-TOF MS-based databases and increasing species differentiation, the use of tandem mass spectrometry-based proteomics has been employed²⁷⁻³⁰, for instance in the discovery of biomarkers of Neisseria meningitidis³⁰.”

Reviewer #2 (Comments for the Author):

Spectrum03542-25, "High-Resolution Differentiation and Identification of the Clinically-Relevant pathogens *Haemophilus influenzae* and *Haemophilus aegyptius*: Combined Whole-Cell MALDI-TOF MS and NanoLC-MS/MS Proteotyping for Defining Protein Biomarkers" by Hamidi et al.

In this work, Hamidi and colleagues describe an approach to extending the MALDI-ToF mass spectrometry database with the aim of improving the identification of species from the genus *Haemophilus*, specifically for discriminating between the closely related species of *Haemophilus influenzae* and *Haemophilus aegyptius*.

A variety of analytical techniques were employed to characterize isolates from both *Haemophilus* species, including whole cell MALDI-ToF MS, tandem nanoLC-MS and comparative genomics. The aim was to generate high-quality MALDI-ToF mass spectra that could be used to reliably differentiate between the two species and identify them. High-resolution tandem LC-MS and genomic analyses were then performed to detect discriminatory, or identifying biomarkers and improve the understanding of the molecular basis for identification and discrimination.

Overall, the study was conducted carefully and is principally relevant due to the varying clinical and veterinary significance of the characterized *Haemophilus* species. The data and results provided are interesting and reliable, and the mass spectrometry techniques used are state of the art. I have no doubts about the validity of the presented data and of most of the results. However, I must say that the conclusions are not always supported by the data and that the way of presenting data and results requires major improvements. While some relevant aspects are only described in part, other points are often listed redundantly in the text.

The manuscript would benefit from a more streamlined text, improved structure and a clearer focus on the work performed.

Author response:

*We have restructured the introduction and discussion, to reduce the overlap between the two sections. Furthermore, we have added sentences describing the focus on the work performed, especially to aim of differentiating *H. influenzae* from *H. aegyptius*.*

Collecting additional data seems to be required (see below)

Furthermore, I would have liked the MALDI spectra of *Haemophilus* species mentioned in the manuscripts to be made available for the review process, at least. In

this era of open data, it is important that the data acquired in the course of the study should be available for verification of the study results.

Author response:

*We have uploaded all the MS data on the MassIVE partner repository with the dataset identifier Title: Proteotyping-Based Differentiation of *H. influenzae* and *H. aegyptius* and MassIVE ID: MSV000099548.*

FTP link: ftp://MSV000099548@massive-ftp.ucsd.edu

We provided the username and the password of the project are facilitated for the reviewing process in the cover letter:

Username: hamideh_hamidi

Password: ccug#bright

Figures and tables: I believe that a detailed list of the microbial isolates examined should be included at the beginning of the manuscript (see Table 1). In my opinion, Table 1 (theoretical masses of *H. influenzae* for internal calibration) is not particularly important in the current version and could therefore be moved to the supplementary information (SI).

Author response:

We agree with the reviewer. We have added a new Table 1 describing the microbial isolates included in the study. Hence, we have moved the Theoretical masses for internal calibration to Supplementary information.

My concerns, comments and suggestions regarding important aspects of the investigations are detailed below.

1. My main criticism is that only a very small number of *H. influenzae* and *H. aegyptius* strains were examined (six *H. influenzae* and three *H. aegyptius*). In my opinion, it is not possible to draw robust conclusions from the examination of such a small number of strains. I believe that independent validation of the results would be necessary. At the very least, additional strains of both species (ideally $n > 10$) should be characterized according to a MALDI standard protocol.

The recorded spectra would then need to be used to verify whether the discriminatory markers postulated in the manuscript were found in the independent samples (i.e. independent validation).

Author response:

We thank the reviewer for this comment and fully acknowledge the concern regarding the limited number of strains analyzed. However, it is important to note that the global availability of Haemophilus aegyptius strains is extremely limited. In the entire CCUG collection, only three H. aegyptius strains are available. Similarly, the DSMZ collection lists only the type strain, and the NCTC collection is also essentially restricted to type strains. No H. aegyptius strains are available in the LMG collection.

In our analysis, based on the NCBI database, only six strains could be clearly defined as H. aegyptius, while nine strains corresponded to H. influenzae biotype aegyptius. Expanding the study to include additional strains is therefore not currently feasible due to the very limited representation or correct identification of H. aegyptius in public culture collections. Moreover, analyzing all available H. influenzae strains (e.g., approximately 163 strains in the CCUG collection) based on whole-genome data would substantially increase costs (in the range of 20,000-30,000 Euros), while the likelihood of identifying additional H. aegyptius strains would remain low (reflected by the low 2% presence in the NCBI database compared to H. influenzae strains).

We agree that independent validation using a larger number of strains would strengthen the conclusions and would be highly desirable. Nevertheless, we believe that our results realistically reflect the current limitations in strain availability and provide a solid foundation for future studies as additional isolates become available.

2. To determine whether the described biomarkers are specific to the respective species, the study should also include other closely and distantly related species (e.g. *H. parainfluenzae*, *H. quentini* and *H. haemolyticus* ssp. *intermedius*, see [5]). This is the only way to determine whether the biomarkers are truly identifying the species or if they are also found in other species of the genus.

Author response:

The primary objective of this study is to differentiate H. influenzae and H. aegyptius, and therefore the specificity of the proposed biomarkers was evaluated exclusively in the context of these two taxa. From a genomic perspective, H. influenzae and H. aegyptius are widely considered to belong to the same species, and the differences observed between them are thus interpreted as intra-species variation rather than inter-species divergence [1].

The genus Haemophilus is characterized by extensive gene sharing and a high frequency of horizontal gene transfer [2,3]. Consequently, the inclusion of additional Haemophilus species could introduce substantial background noise and potentially confound the identification of biomarkers that are relevant for discriminating between these two closely related taxa [3]. While the analysis of other species may be informative for assessing specificity at the genus level, this falls outside the scope and aims of the present study.

Moreover, *H. parainfluenzae* can be differentiated from *H. influenzae* based on biochemical traits and growth factor requirements, as it does not require X factor. Phenotypic discrimination of *H. parainfluenzae* from other *Haemophilus* and *Aggregatibacter* species is usually unproblematic, and it forms a distinct phylogenetic cluster based on 16S rRNA gene and whole-genome analyses [4]. “*H. quentini*” is not validly published [5], however it can be differentiated from *H. influenzae* by sequencing of the 16S rRNA gene or housekeeping genes and therefore represents a cryptic genospecies [4].

H. haemolyticus is readily distinguished from encapsulated *H. influenzae*, but discrimination from nonencapsulated strains is challenging due to high phenotypic and genetic similarity. PCR assays targeting *hpd* and *fucK*, along with 16S rRNA analysis, enable reliable discrimination, combining phenotypic and genotypic approaches improves accuracy. Notably, “*H. haemolyticus* subsp. *Intermedius*” is not validly named and has no standing in nomenclature [4,6].

Based on these considerations, restricting the analysis to *H. influenzae* and *H. aegyptius* is appropriate and justified given the diagnostic focus of this work.

[1] Nørskov-Lauritsen, N. (2021). *Haemophilus*†,‡. In *Bergey's Manual of Systematics of Archaea and Bacteria* (eds M.E. Trujillo, S. Dedysh, P. DeVos, B. Hedlund, P. Kämpfer, F.A. Rainey and W.B. Whitman). <https://doi.org/10.1002/9781118960608.gbm01198.pub2>

[2]Takahata S, Ida T, Senju N, Sanbongi Y, Miyata A, Maebashi K, Hoshiko S. Horizontal gene transfer of *ftsI*, encoding penicillin-binding protein 3, in *Haemophilus influenzae*. *Antimicrob Agents Chemother.* 2007 May;51(5):1589-95. doi: 10.1128/AAC.01545-06. Epub 2007 Feb 26. PMID: 17325223; PMCID: PMC1855551.

[3] Maughan H, Redfield RJ. Extensive variation in natural competence in *Haemophilus influenzae*. *Evolution.* 2009 Jul;63(7):1852-66. doi: 10.1111/j.1558-5646.2009.00658.x. Epub 2009 Feb 23. PMID: 19239488.

[4] Nørskov-Lauritsen N. 2014. Classification, Identification, and Clinical Significance of *Haemophilus* and *Aggregatibacter* Species with Host Specificity for Humans. *Clin Microbiol Rev* 27:..<https://doi.org/10.1128/cmr.00103-13>

[5] Glover, W. A., Suarez, C. J., & Clarridge III, J. E. (2011). Genotypic and phenotypic characterization and clinical significance of ‘*Haemophilus quentini*’ isolated from the urinary tract of adult men. *Journal of medical microbiology*, 60(11), 1689-1692.

[6] Anderson R, Wang X, Briere EC, Katz LS, Cohn AC, Clark TA, Messonnier NE, Mayer LW. *Haemophilus haemolyticus* isolates causing clinical disease. *J Clin Microbiol.* 2012 Jul;50(7):2462-5. doi: 10.1128/JCM.06575-11. Epub 2012 May 9. PMID: 22573587; PMCID: PMC3405640.

3. Section MALDI-ToF mass spectrometry, page 4 lines 126-135. MALDI-ToF mass spectrometry of bacteria is a widely employed technique with thousands of published studies. Against this background, I find it incomprehensible that new methods of sample preparation and for MALDI measurements were established and optimized instead of using one of the standard methods.

While using alternative methods is certainly possible, such methods must be then documented in much greater level of detail. However, this section description

contains only a limited amount of information, and I believe that the methodology is not comprehensible to third parties.

Author response:

We noticed that we did not refer correctly in the manuscript to the extensive explanation about the sample preparation approach we employed in the manuscript. We have included this in the Supplementary information 1. Since we are expanding the m/z range of the analysis, targeting additional high-molecular weight protein biomarkers, instead of the commonly used ribosomal markers in the m/z 2,000-20,000 range, hence enabling intra-species differentiation. Thus, we were not able to use common sample preparation practices, instead we needed to optimize it for this particular analysis.

4. Page 8, line 153. What is a MALDI Biotyper smart instrument? Bruker does not offer such an instrument. Please correct.

Author response:

We apologize for any confusion. The instrument used in this study was the MALDI Biotyper SMART RUO system, which includes both the MALDI-TOF mass spectrometer (microflex LT system) and the associated SMART MBT software and reference library for research use only. To ensure clarity, we adjusted the Materials and Methods section to the specific MALDI Biotyper microflex LT system, which more accurately describes the instrument used, page 9 line 186-188.

“The prepared target was then allowed to dry at room temperature before conducting MALDI-TOF MS measurements using the MALDI-TOF MS Biotyper Microflex LT System and the SMART MBT software and reference library for research use only (Bruker Daltonics, Germany).”

5. Section NanoLC-MS/MS Analysis of Proteotyping Samples Page 9: (As a comment only.) These measurements were performed in DDA mode. DIA mode acquisition schemes offer much more complete proteomic information, so it is unclear why this somewhat outdated methodology was used.

Author response:

We thank the reviewer for raising the important point about acquisition mode. While we acknowledge that DIA has gained popularity and offers advantages in completeness and reproducibility, it is not accurate to characterize data-dependent acquisition (DDA) as outdated. Both DDA and DIA continue to be widely used in contemporary quantitative proteomics, each with distinct strengths and appropriate use cases.

DDA remains a robust and extensively validated approach for peptide discovery, identification, and quantitation. Many recent high-impact proteomics studies,

including single-cell and clinical applications, still utilize DDA workflows because of their compatibility with spectral library generation and well-established analysis pipelines, particularly in discovery proteomics and structural characterization. Indeed, current quantitative proteomics publications in 2024 and 2025 continue to feature DDA, especially in studies integrating isobaric labeling (e.g., TMT/iTRAQ) or in workflows where DDA-based spectral libraries support downstream DIA analyses.

In recent years, DIA has been highlighted as a powerful alternative due to its enhanced data completeness, reduced missing values, and improved reproducibility. Comparative studies have shown that DIA can identify a greater number of proteins with lower technical variation than DDA in specific biological matrices and DIA workflows and informatics tools are indeed expanding rapidly in quantitative proteomics.

The choice of DDA in our study reflects the experimental design and goals, including established informatics pipelines and compatibility with existing spectral resources. DDA provides high specificity, well-annotated fragmentation spectra, and remains appropriate for proteome discovery in complex biological samples. The reviewer's point regarding completeness is well taken, and we agree that DIA offers advantages in quantitative coverage; however, this does not render DDA obsolete. Both approaches remain scientifically valid and are frequently used in tandem or in complementary ways in the current proteomics literature. In summary, while DIA continues to grow in adoption and offers advantages in certain contexts, DDA is not outdated but rather remains a foundational acquisition strategy in quantitative proteomics and spectral library generation.

6. It is not clear to me how and why spectra were transferred from the Bruker MBT ecosystem to the SARAMIS software from bioMérieux. Firstly, it would be interesting to know how exactly the spectral data recorded on a Bruker Daltonics mass spectrometer was imported into the SARAMIS software. I also find it difficult to understand why the Bruker software (MBT) was not used for the analysis. The logical approach for this study would be to first use MBT and the RUO database to identify the recorded data, and then expand the existing databases to include MSPs from *Haemophilus influenzae* and *Haemophilus aegyptius*. Subsequently, identification of additional, previously unused strains should be based on the expanded, customer-specific database. It is unclear why the authors did not take this approach, which has already been used in many other studies (see also my previous comments).

Author response:

We thank the reviewer for this valuable comment and agree that, in our study, MALDI-TOF spectra were acquired on a Bruker Daltonics instrument. The raw data were converted to mzXML format using the Bruker CompassXport utility, which enabled import into the SARAMIS (bioMérieux) software environment. The rationale

for using SARAMIS was not primary identification, but rather its strength in biomarker-based analysis and categorization of discriminatory peaks. Specifically, SARAMIS was used to identify and classify biomarker peaks into three categories: (i) biomarkers conserved at the genus or higher taxonomic level, (ii) biomarkers specific to either Haemophilus influenzae or Haemophilus aegyptius, and (iii) biomarkers shared by all strains of both species. We would also like to clarify that the reviewer's suggested approach was, in fact, initially applied. The acquired spectra were first searched against the Bruker MBT RUO library, where H. influenzae and H. aegyptius were consistently classified as closely related species, highlighting the known limitations of standard protein-based identification for their discrimination. Based on these results, and in line with the study objective, we explored alternative strategies to improve differentiation. To this end, we extended the analyzed range up to m/z 40,000 (beyond the conventional lower mass range used for routine MBT identification). This approach enabled the detection of high-intensity, species-informative peaks that are not typically exploited in standard MBT workflows. As described in the manuscript, these biomarkers provided improved discriminatory power between H. influenzae and H. aegyptius.

The following paragraph was added in page 10, line 210-216.

“The raw data were converted to mzXML format using the Bruker CompassXport utility, which enabled import into the SARAMIS (bioMérieux, Marcy-l'Étoile, France) software environment. The rationale for using SARAMIS was not primary identification, but rather its strength in biomarker-based analysis and categorization of discriminatory peaks. Specifically, SARAMIS was used to identify and classify biomarker peaks into three categories: (i) biomarkers conserved at the genus or higher taxonomic level, (ii) biomarkers specific to either H. influenzae or H. aegyptius, and (iii) biomarkers shared by all strains of both species.”

7. The methodological approach of combining MALDI-ToF MS and tandem LC-MS to increase the understanding of the molecular basis, as well as the accuracy of identification and differentiation, is adequate but not particularly novel. Other authors have previously used this and similar approaches in earlier studies to improve understanding of MALDI-based diagnostics, e.g. for biomarker discovery and validation. Therefore, I think it would be fair to cite studies such as Fagerquist (2016) Expert Rev Proteomics [1], Lasch et al. (2016) TRAC [2], and Dieckmann et al. (2008) AEM [3], perhaps also ref. [4]. To avoid excessive citation, some redundant citations in the 'Introduction' section could be omitted (lines 62–63, page 4, citations #1–#6).

Author response:

We agree with the Reviewer and we have added this reference to the introduction at page 6 line 126-130. We did not, however, remove any references from the existing list, since we wanted to keep the ones we had.

“To improve MALDI-TOF MS-based databases and increasing species differentiation, the use of tandem mass spectrometry-based proteomics has been employed²⁷⁻³⁰, for instance in the discovery of biomarkers of Neisseria meningitidis³⁰.”

8. There is some overlap in the statements in the two sections, 'Abstract' and 'Importance'. Perhaps the authors could take another critical look at both paragraphs and try to reduce the redundancy of their statements.

Author response:

We have rewritten the Importance section accordingly page 3 line 47-63.

“Species-level resolution of clinically relevant pathogens remains a significant challenge in clinical diagnostics. Among Haemophilus species, H. influenzae and H. aegyptius are particularly problematic because of their high genetic similarity, overlapping phenotypes, and distinct clinical implications. Conventional techniques often fail to reliably distinguish between these two pathogens, frequently misclassifying them. While diagnostic challenges are well recognized within the H. influenzae species complex, effective differentiation between H. influenzae and H. aegyptius remains comparatively underexplored. Accurate and rapid identification is critical, as these opportunistic pathogens differ in disease manifestations and epidemiology. Limitations in differentiating these closely related species can lead to misidentification and suboptimal treatment strategies. Our study presents a robust and reproducible high-resolution nanoLC-MS/MS proteotyping approach, whole-cell MALDI-TOF MS, and comparative genomics, which enables confident taxonomic classification of H. influenzae and H. aegyptius species down to the species and strain levels. By identifying species-unique peptide biomarkers, our method addresses the diagnostic gap left by standard MALDI-TOF MS and genomic approaches, offering an advanced tool with direct clinical relevance. This has important implications for improving species-level microbial diagnostics, understanding species-specific pathogenicity, and supporting accurate epidemiological surveillance.”

9. Pages 12 and 13, section 'Results and Discussion': This long section reads like a second introduction. I suggest significantly shortening this section and moving it to the introduction, as it contains redundant information. The results section should start with a description of the results.

Author response:

We agree with the reviewer and we have removed the section on page 12 and 13. Instead we start with the following statement at the start of the Results and Discussion section

“In this study, we combined high-resolution nanoLC-MS/MS proteotyping with whole-cell MALDI-TOF MS and comparative genomics to identify 31 distinct protein biomarkers uniquely and consistently associated with Haemophilus influenzae and H. aegyptius. These biomarkers demonstrate strong potential for accurate species-level identification. Genomic validation further confirmed their specificity and reliability as diagnostic markers.”

We moved one part of this cut-out section into the introduction

“Haemophilus species are Gram-negative bacteria that commonly inhabit the respiratory tracts and mucosal surfaces of humans and animals. The Haemophilus genus includes fourteen recognized and validly published species, with H. influenzae being the type and most clinically significant species of the genus¹⁰. Haemophilus influenzae and Haemophilus aegyptius are frequently encountered in clinical microbiology laboratories and exhibit a range of pathogenicity, from non-pathogenic commensals to highly pathogenic strains responsible for severe conditions such as respiratory tract infections (RTIs), conjunctivitis, and life-threatening invasive diseases like septicemia, sepsis, and meningitis. Various protocols have been developed to distinguish H. influenzae from closely related species of lesser pathogenic importance. However, accurate identification based on phenotypic characteristics remains challenging and often unreliable²⁹. DNA-DNA hybridization analysis suggests that H. influenzae and H. aegyptius may not be distinct species³⁰, and MLSA places them within the core of the species^{31,32}. Despite this, H. influenzae and H. aegyptius have traditionally been classified as distinct species. H. aegyptius is mainly associated with conjunctivitis and other ocular infections. Clinical diagnosis typically depends on observing symptoms such as eye redness and discharge, which are confirmed through laboratory testing. In some cases, infections caused by H. aegyptius have been linked to more serious conditions, like purpuric fever, which can be fatal³³. Conversely, H. influenzae is associated with a variety of infections, including respiratory tract infections, meningitis, and epiglottitis. The differing clinical presentations of these two species underscore the importance of accurate identification to guide appropriate treatment and management.”

10. Page 18, In the strict sense, ANI values larger than 95% (sometimes 96% are stated) do not support the definition of *H. influenzae* and *H. aegyptius* as separate species. Genomes sharing an ANI value above this threshold are generally considered to belong to the same species, while those with values below 90% are typically considered different species. This could be mentioned in somewhere in the manuscript.

Author response:

The comment suggested by the reviewer has been addressed, and the sentence “, ANI values of $\geq 94\sim 96\%$ were considered indicative of strains belonging to the same

species, whereas values below this range were regarded as representing different species, according to the established species delineation criteria.^{36,38} has been added in lines 391–393.

11. Page 20, line 440. Please complete this sentence.

Author response:

The sentence in line 437 has been completed with the following text: “the role of the selected proteins as reliable biomarkers for differentiating these closely related species.”

[1] <http://www.ncbi.nlm.nih.gov/pubmed/27838927>

[2] <http://dx.doi.org/10.1016/j.trac.2016.04.013>

[3] <http://www.ncbi.nlm.nih.gov/pubmed/18952875>

[4] <http://www.ncbi.nlm.nih.gov/pubmed/23916798>

[5] <https://www.ncbi.nlm.nih.gov/pubmed/35139905>

Re: Spectrum03542-25R1 (High-Resolution Differentiation and Identification of the Clinically-Relevant pathogens Haemophilus influenzae and Haemophilus aegyptius: Combined Whole-Cell MALDI-TOF MS and NanoLC-MS/MS Proteotyping for Defining Protein Biomarkers)

Dear Dr. Roger Karlsson:

Your manuscript has been accepted, and I am forwarding it to the ASM production staff for publication. Your paper will first be checked to make sure all elements meet the technical requirements. ASM staff will contact you if anything needs to be revised before copyediting and production can begin. Otherwise, you will be notified when your proofs are ready to be viewed.

Sincerely,
Takeaki Wajima
Editor
Microbiology Spectrum

Reviewer #1 (Comments for the Author):

The authors answered appropriately to the reviewers' comments.